



# Subgrid-scale Horizontal and Vertical Variations of Cloud Water in Stratocumulus Clouds: A case study based on LES and comparisons with in-situ observations

Justin A. Covert[1], David B. Mechem[1], and Zhibo Zhang[2,3]

[1]Department of Geography and Atmospheric Science, University of Kansas, Lawrence, KS, United States
[2]Joint Center for Earth Systems Technology, UMBC, Baltimore, MD, United States
[3]Department of Physics, UMBC, Baltimore, MD, United States

**Correspondence:** Justin A. Covert (jacovert@ku.edu)

**Abstract.** Stratocumulus clouds in the marine boundary layer cover a large fraction of ocean surface and play an important role in the radiative energy balance of the Earth system. Simulating these clouds in Earth system models (ESMs) has proven to be extremely challenging, in part because cloud microphysical processes such as the autoconversion of cloud water into precipitation occur at the scales much smaller than typical ESM grid sizes. An accurate autoconversion parameterization needs to

account for not only the local microphysical process (e.g., the dependence on cloud water content $q_c$ and cloud droplet number concentration $N_c$), but also the sub-grid scale variability of the cloud properties the determine the process rate. Accounting for subgrid-scale variability is often achieved by the introduction of a so-called enhancement factor $E$. Previous studies of $E$ for autoconversion focused more on its dependence on cloud regime and ESM grid size, but largely overlooked the vertical dependence of $E$ within the cloud. In this study, we use a large-eddy simulation (LES) model, initialized and constrained with

in situ and surface-based measurements from a recent airborne field campaign, to characterize the vertical dependence of the horizontal variation of $q_c$ in stratocumulus clouds and the implications for $E$. Similar to our recent observational study (Zhang et al., 2021), we found that the inverse relative variance of $q_c$, an index of horizontal homogeneity, generally increases from cloud base upward through the lower 2/3 of the cloud, and then decreases in the uppermost 1/3 of the cloud. As a result, $E$ decreases from cloud base upward, and then increases towards the cloud top. We apply a decomposition analysis to the LES

cloud water field to understand the relative roles of the mean and variances of $q_c$ in determining the vertical dependence of $E$. Our analysis reveals that the vertical dependence of the horizontal $q_c$ variability and enhancement factor $E$ is a combined result of condensation growth throughout the lower portion of the cloud and entrainment mixing at cloud top. The findings from this study indicate that a vertically dependent $E$ should be used in ESM autoconversion parameterizations.

## 1   Introduction

Marine boundary layer (MBL) clouds play an important role in Earth's climate system. Stratocumulus clouds are one of the predominant types of MBL cloud systems, covering more area in annual mean (∼20% of the Earth's surface) than other MBL clouds (Warren et al., 1986; Wood, 2012). Between their high albedo and large temporal and areal coverage, MBL stratocu-





mulus significantly influence Earth's radiative budget (Klein and Hartmann, 1993; Martin et al., 1995; Ghate et al., 2014). As such, a realistic representation of MBL stratocumulus clouds in Earth system models (ESMs) and a faithful representation of

their interactions with the climate system is extremely important for predicting and understanding future climate (Cess et al., 1990; Bony et al., 2015). However, representing these clouds in ESMs is extremely challenging because many of the physical processes in MBL clouds occur at scales much smaller than the grid size of current ESMs. The scale of most ESMs is tens to hundreds of kilometers, whereas cloud microphysical processes occur at scales of tens of meters or smaller (Randall et al., 2003). Hence, most sub-grid scale cloud-physics processes must be parameterized using ESM-resolvable variables.

Most precipitation falling from frontal or deep convective clouds involves ice-phase processes, whereas precipitation from MBL stratocumulus generally forms at temperatures above $0°C$ (or at least above the temperature where appreciable ice nucleation takes places) through the collision and coalescence of cloud droplets known as the warm rain processes (Pruppacher and Klett, 1997). Precipitation exerts both obvious and subtle effects on stratocumulus cloud properties, specifically the vertical distribution of water, MBL stability, and the magnitude of turbulent fluxes and entrainment (Stevens et al., 1998). These effects

can lead to cloud lifetime effects because they substantially influence cloud radiative properties through both changes to cloud cover and the radiative properties of the clouds themselves (Albrecht, 1989).

Because of the profound influence of precipitation processes on the radiative properties of stratocumulus, they must be accurately parameterized within ESMs (Pawlowska and Brenguier, 2003; Wood, 2005; Mülmenstädt et al., 2020). Many current ESMs use two-moment bulk microphysics schemes to simulate cloud microphysics (e.g., Morrison and Gettelman, 2008). In

this type of scheme, the full drop size distribution is separated into cloud and rain particle size distributions based on a separation size $r_0$. Both distributions are characterized by mass (liquid water mixing ratio) and number concentration. Accordingly, the collision-coalescence processes are represented by autoconversion and accretion. Autoconversion is the process of cloud drops colliding with one another to form rain drops ($r > r_0$), and accretion is the collection of cloud droplets ($r < r_0$) by rain drops (Wood, 2005). Although many different types of autoconversion parameterization schemes have been developed,

autoconversion is typically parameterized within many ESMs as a power function of liquid cloud water $q_c$ and cloud droplet number concentration $N_c$. One commonly used scheme developed by Khairoutdinov and Kogan (2000) parameterizes the autoconversion rate as

$$\left(\frac{\partial q_r}{\partial t}\right)_{auto} = C q_c^{\beta_q} N_c^{\beta_N} \tag{1}$$

in which $q_r$ is the rain water content and the parameters C = 1350, $\beta_q$ = 2.47, and $\beta_N$ = -1.79 are derived from least-squares

nonlinear regression of drop size spectra from bin microphysics large-eddy simulation (LES) output. In (1), $q_c$ is the cloud water mixing ratio in kg kg$^{-1}$ and $N_c$ is the cloud droplet concentration in cm$^{-3}$. A parameterization with this functional form leads to a highly non-linear dependence of the autoconversion rate on the $q_c$.

Although a microphysical parameterization scheme such as Khairoutdinov and Kogan (2000) is necessary for the simulation of warm-rain processes in ESMs, it represents a local process rate and by itself is not sufficient for determining autoconver-

sion across an ESM grid volume. As previously mentioned, cloud processes generally occur at scales (e.g., tens of meters and smaller) much smaller than the typical grid size of EMSs (e.g., $O(100$ km)), and these sub-grid variations of cloud properties



must also be accounted for in ESM parameterizations. The effect of subgrid-scale variability on determining mean micro-physical process rates over a model grid volume has has been discussed in several previous studies (Pincus and Klein, 2000; Larson et al., 2001; Zhang et al., 2019). Essentially, for a microphysical process rate $f(x)$ such as autoconversion, which is a

function of a cloud property $x$, its grid-mean value $\langle f(x) \rangle$ should be computed as $\langle f(x) \rangle = \int f(x)P(x)dx$, where $P(x)$ is the probability density function (PDF) of $x$ over the grid volume. However, because most ESMs lack information of sub-grid cloud variation, $P(x)$ is unknown and the grid-mean $\langle f(x) \rangle$ can only be estimated from the grid-mean value of the cloud property, i.e., $f(\langle x \rangle)$. For a linear process, $f(\langle x \rangle) = \langle f(x) \rangle$. But, according to Jensen's inequality, calculating a non-linear processes such as autoconversion and accretion using the grid-mean average of $x$ (e.g., $q_c$) is not equal to the process rate calculated

everywhere over the grid volume and then averaged. That is, $\langle f(x) \rangle \neq f(\langle x \rangle)$.

To account this bias arising from neglecting subgrid-scale variability in our estimation of $\langle f(x) \rangle$, a so-called "enhancement factor" $E$ is introduced so that $\langle f(x) \rangle = E \cdot f(\langle x \rangle)$. As discussed in several previous studies, the value of the enhancement factor depends primarily on two points. First, it depends on the non-linearity of the function $f(x)$. In general, given the same sub-grid variation of $x$, a more non-linear the function $f(x)$ will yield a larger enhancement factor $E$ (Pincus and Klein, 2000;

Larson and Griffin, 2013). This effect explains why the enhancement factor for the highly nonlinear autoconversion rate is usually larger than that for accretion (Zhang et al., 2019). Second, given a cloud process $f(x)$, enhancement factor $E$ increases with the sub-grid variance of $x$. This is the reason why the enhancement factor for the autoconversion is typically larger for open-ocean cumulus regions where clouds are more in-homogeneous compared to stratocumulus regions where clouds are more homogeneous to (Lebsock et al., 2013; Zhang et al., 2019). It also explains why the enhancement factor usually increases

with ESM grid size, with a larger grid size exhibiting greater variability. Despite this qualitative understanding, $E$ is often assumed as a constant for simplicity due to the inability to simulate sub-grid scale cloud variability. Based on a previous study by Morrison and Gettelman (2008), a widely used value for the enhancement factor for autoconversion in CAM and many ESMs is 3.2.

A number of previous studies have investigated the enhancement factors for autoconversion and other cloud processes using

different approaches. Some studies aimed to understand dependence of enhancement factors on cloud regimes using satellite observations (e.g., Lebsock et al., 2013; Zhang et al., 2019), whereas others investigated the dependence of cloud property variance and enhancement factors on ESM grid-size, hoping to develop so-called scale-aware parameterization schemes (e.g., Boutle et al., 2014; Hill et al., 2015; Xie and Zhang, 2015; Ahlgrimm and Forbes, 2016). Although these studies shed important light on the problem, with few exceptions they share two common limitations. First, they consider only the impacts of sub-grid

$q_c$ variations on the $E$ for autoconversion but ignore the impacts of sub-grid variation of $N_c$ and its covariation with $q_c$. In a recent study based on satellite observations, Zhang et al. (2019) illustrated that in addition to $q_c$ variability, the sub-grid variance of $N_c$ also increases the value of $E$. More importantly, they elucidate that $q_c$ and $N_c$ are often positively correlated in stratocumulus clouds, and the neglect of the covariation between $q_c$ and $N_c$ in the formation of $E$ for autoconversion can lead to significant bias. This finding is confirmed by a latest study by Zhang et al. (2021) (referred to as Z21 hereafter) based

on in situ measurements from the Aerosol and Cloud Experiments in the Eastern North Atlantic (ACE-ENA) campaign. They showed that the correlation coefficient between $q_c$ and $N_c$ can be as high as 0.95 at cloud top, which substantially reduces





the value of $E$ and therefore counteracts the effects of $q_c$ and $N_c$ variations. To distinguish the difference from impacts of $N_c$ variability, following Z21 we shall use "$E_q$" in this study to refer to the impact of sub-grid cloud $q_c$ on autoconversion.

The other common limitation of previous studies is the neglect of the vertical dependence of cloud horizontal variability.
Kogan and Mechem (2014, 2016) showed the importance of the vertical structure of horizontal variability and enhancement factors in shallow cumulus and congestus clouds. Stratocumulus clouds exhibit a distinct vertical structure resulting from a combination of processes such as adiabatic condensational growth, collision-coalescence, and entrainment mixing. As a result, the horizontal variations of cloud properties and therefore $E$ depend on the vertical location inside the clouds. It is important to understand this dependence to better parameterize and simulate warm rain processes in ESMs. Aiming to fill this gap in our
knowledge, Z21 used in situ measurements of MBL clouds from the ACE-ENA campaign to quantify the vertical dependence of the horizontal variations of $q_c$ and $N_c$, as well as their correlation. Z21 first identified the horizontal legs in each research flight and then derived the corresponding horizontal variability and covariability of $q_c$ and $N_c$. Z21 found that the mean value of $q_c$ (i.e., $\langle q_c \rangle$) tends to increase from cloud base upward in a manner consistent with an adiabatic (or near-adiabatic) liquid water lapse rate associated with condensational growth (Zuidema et al., 2005), and peak near cloud top before sharply decreasing
near the inversion (see Fig. 4 in Z21). The horizontal homogeneity of the clouds, which is defined as the ratio of $\langle q_c \rangle^2$ to the variance of $q_c$ (see Eq. (2)), follows a similar pattern that increases first from cloud base upward and peaks below cloud top. As a result of this vertical structure, the enhancement factor $E$ for the autoconversion parameterization due to $q_c$ variation tends to decrease from cloud base toward cloud top, with a minimum value below cloud top, and then increase slightly toward cloud top. This observation-based study shed important light on the vertical dependence of the sub-grid variation of $q_c$ and
the corresponding impacts on $E$ and autoconversion process rates. An important implication is that the value of $E$ has a strongly vertical dependence so the use of a constant value can lead to substantial error in the simulation of autoconversion. However, Z21 also faces two important limitations. First, as an observation-based study, Z21 provided only a phenomenological description of the vertical dependence of the horizontal $q_c$ variation and did not explain the underlying physics, especially the connections of $q_c$ vertical structure to the known cloud processes such as condensation growth and entrainment. Second, the
horizontal legs in each research flight (RF) only sampled 3 to 4 levels in the stratocumulus cloud, providing only a crude picture of cloud vertical structure.

This study is inspired and motivated by Z21. We use a state-of-the-art LES to simulate a stratocumulus cloud case (July 18, 2017) observed during the ACE-ENA campaign. The rich information from LES, in particular the high-resolution cloud profile, allows us to overcome the limitations in Z21 and shed fresh light on the three-dimensional (3D) structure of $q_c$ and the
associated implications for the enhancement factor $E$ for autoconversion. The main objectives of this paper are (1) to use LES to achieve a process-level understanding of the vertical dependence of the horizontal variation of cloud water in stratocumulus clouds, in particular the connections with key microphysical processes such as condensational growth and entrainment, and (2) better understand how these sub-grid scale variations of cloud water affect the enhancement factor associated with warm-rain processes in ESMs. Z21 identify the importance of variability in both $q_c$ and $N_c$, as well as covariability between them, but here
we first address only variability in $q_c$, with $N_c$ being a fixed parameter. Subsequent work will broaden the scope to consider both $q_c$ and $N_c$ interactively. The paper is organized as follows. In section 2 we summarize the the ACE-ENA campaign and



field observations relevant for this study. Section 3 describes the LES model and setup of the simulation suite, and Sec. 4 discusses the simulation results. Our conclusions are summarized in Sec. 5.

## 2 July 18 2017 case study from the ACE-ENA campaign

The ACE-ENA campaign coordinated by the Department of Energy (DOE) Atmospheric Radiation Measurement (ARM) program was aimed to obtain comprehensive in situ characterizations of MBL structure and associated vertical distributions and horizontal variability of low-cloud and aerosol properties in the vicinity of the East North Atlantic (ENA) ARM site on Graciosa Island in the Azores archipelago (Wang et al., 2021). The campaign included two intensive measurement periods (IOPs), one in the summer of 2017 from June 21 to July 20, and the other in the following winter from January 15 to February

18, 2018. The ARM Aerial Facility (AAF) Gulfstream-1 (G-1) aircraft was deployed for 39 research flights (RF) during the two IOPs around the ARM ENA site that sampled a large variety of cloud and aerosol properties along with the meteorological conditions.

Of particular interest in this study is the July 17, 2017 RF, the same "golden case" studied in Z21. On this day, the North Atlantic was characterized by a low pressure system to the north and the Azores high to the south. The ARM ENA site was at

the southern tip of the cold-air sector of low pressure system behind the cold front (Kazemirad and Miller, 2020). As a result, the region is mostly covered by fair-weather low-level stratocumulus clouds. Fig. 1a shows the vertical track of the G-1 aircraft overlaid on the reflectivity curtain of ground based Ka–band ARM zenith cloud radar (KAZR, ARM, 2019). In this RF, the G1 aircraft performed a number legs at constant altitude (referred to as "hlegs") in a "V" shape at different vertical levels inside the stratocumulus clouds. Z21 identified 14 hlegs in this RF and selected 7 (hleg 5-8 and 10-12 in Fig. 1a) to study the horizontal

variations of $q_c$ and $N_c$. Each of this selected hlegs constitutes a horizontal sampling of the stratocumulus clouds at the scale of about 30 km, which can be considered as a "virtual" ESM grid. The 10-Hz in situ measurements provide the small-scale variations of $q_c$ and $N_c$ within the $\sim$30 km ESM grid, which are used in the Z21 to derive, for example, the variances of $q_c$ and the corresponding $E$ for autoconversion. Note that these hlegs are located at different vertical levels. For example, hleg 5 and 8 are at cloud base and top, respectively. Therefore, together they provide an excellent set of samples of the MBL cloud

properties at different vertical levels that are used in Z21 to study the vertical dependence of horizontal $q_c$ and $N_c$ variations.

Figure 1b shows the vertical dependence of the inverse relative variance (IRV) of $q_c$ calculated from the 7 selected hlegs. The IRV $\nu_{q_c}$ is defined as follows

$$\nu_{q_c} = \frac{\langle q_c \rangle^2}{\text{Var}(q_c)} \tag{2}$$

where $\langle q_c \rangle$ and $\text{Var}(q_c)$ are the mean and variance of $q_c$, respectively. As such, the IRV can be considered as an index of cloud

water horizontal homogeneity where a more homogeneous cloud water field will have a larger value of IRV. As shown in Fig. 1b the value of $\nu_{q_c}$ is only about 1.0-1.25 at cloud base (i.e., hlegs 5 and 10). It first increases upward for the hlegs in the center of the cloud (i.e., hlegs 6, 7 and 11) and peaks at about 1 km for the hleg 7. Interestingly, the value of $\nu_{q_c}$ for the two cloud top hlegs (8 and 12) becomes smaller than that of hleg 7. Because the value of $E$ for autoconversion due to $q_c$ variation is inversely





proportional to $\nu_{q_c}$, $E$ is seen to first decrease from cloud base upward until it peaks at around 1 km (i.e., hleg 7), and then increase slightly toward cloud top. This result indicates that the $E$ should not be simply treated as a constant independent of vertical location in the cloud.

As explained earlier, the Z21 study faces important limitations. In particular, it provides only a phenomenological description of the vertical dependence of the horizontal $q_c$ variation and did not explain the underlying physics, particularly the connections of $q_c$ vertical structure to the processes such as condensational growth, collision–coalescence, and entrainment, known to govern stratocumulus behavior. This lack of process-level understanding and the limitation of in situ measurements (i.e., coarse vertical sampling rate) make it difficult to assess to what extent the vertical structure of $\nu_{q_c}$ and $E$ revealed in the July 18, 2017 case is representative of MBL stratocumulus clouds. These limitations motivated our LES study, which aims to achieve a process-level understanding of the vertical structure of $\nu_{q_c}$ and $E$.

## 3 LES model description and configuration

The LES model used in this study is the System for Atmospheric Modeling (SAM, Khairoutdinov and Randall, 2003). SAM is based on a non-hydrostatic and anelastic equation set. The momentum variables use 2nd-order centered spatial differences and third-order Adams-Bashforth time differencing. Scalar advection uses the fifth-order advection scheme (ULTIMATE–MACHO, Yamaguchi et al., 2011), which minimizes artificial numerical diffusion across the inversion. Subgrid-scale fluxes are formulated using the 1.5-order approach of Deardorff (1980). Horizontal boundary conditions are doubly periodic. The upper boundary is a rigid lid with Rayleigh damping applied to the upper 1/3 of the domain to minimize reflection of internal gravity waves off the top boundary.

The control simulation employs a domain size of $30.24 \times 30.24 \times 20$ km$^3$ ($864 \times 864 \times 192$ grid points), a horizontal scale roughly similar to the 7 selected hlegs in Z21. All additional sensitivity simulations employ a smaller domain size of $8.96 \times 8.96 \times 20$km$^3$ ($256 \times 256 \times 192$ grid points). The horizontal grid spacing is 35 m while the vertical grid spacing is 5 m near the surface and inversion layer, increasing to near 1500 m at the top of the domain. The deep domain is required to accurately calculate the downwelling radiation streams at cloud level. Shortwave and longwave fluxes are calculated using the radiation parameterization adapted from the NCAR Community Climate Model (CCM3, Kiehl et al., 1998).

Microphysical processes are represented using a simplified version of the Khairoutdinov and Kogan (2000) parameterization. Similar to the classic Kessler parameterization (Kessler, 1969), Khairoutdinov and Kogan is based on partial moments of the drop size distribution. As originally formulated, it is a fully interactive, 2-moment parameterization that includes conservation equations for mass and number concentration of both cloud (i.e., $q_c$ and $N_c$) and rain (i.e., $q_r$ and $N_r$ where the subscript "r" indicates the rain) species. In this study, the parameterization is simplified somewhat by holding the droplet concentration fixed at a user-specified value. In this approach, specifying the $N_c$ is a proxy for the cloud concentration nuclei (CCN) concentration. Our control simulation uses a value of $N_c = 75$ cm$^{-3}$, which is based on airborne in situ measurementd of the July 18, 2017 case (Zhang et al., 2021). In addition to the control simulation, several model sensitivity runs are performed to characterize the impacts of varying $N_c$ between 50 and 100 cm$^{-3}$ (see section 4.4). For reasons of computational expense, these sensitivity



simulations are run over smaller domains of $256{\times}256{\times}192$. Fixing the $N_c$ value allows us to isolate how the variability in liquid water influences microphysical process rates. Future studies will include interactive $N_c$ values to explore the mutual influence of the variability of liquid water and droplet concentration on process rates.

Model initial conditions are based on the 1138 UTC sounding from Graciosa Island, roughly coinciding with the G1 flight and our main analysis interval from 0900 UTC to 1200 UTC. The soundings before and after the 1138 UTC soundings indicate a deepening of the boundary layer followed by a reduction of depth (Fig. 2). The moisture profile seems to better represent the inversion structure compared with the potential temperature structure, which is more diffuse, a behavior commonly observed in stratocumulus-topped boundary layers (Caldwell et al., 2005). The initial LES mean profiles are constructed from simple

piece-wise approximations to the sounding profiles. Mixed layer $\theta_l$ and $q_t$ are 292.2 K and 11.2 g kg$^{-1}$, respectively, with jumps across the inversion of $\sim$7 K and $\sim$–3 g kg$^{-1}$. The height of the inversion in the LES initial conditions is 1132.5 m (895 hPa).

Model simulations run from 0600 UTC to 1500 UTC for 18 July 2017. Large-scale vertical velocity and advective tendencies of temperature and moisture are provided from the 3-hourly DOE ARM variational analysis product (VARANAL, Zhang and

Lin, 1997; Xie et al., 2014) and are shown in Fig. 3. The simulation period is characterized by cold advection throughout the depth of the troposphere. Drying at low levels is overlaid by a layer that is drying from 0300-0900 UTC and then moistening from then onward. The vertical motion field is dominated by subsidence. Although we could calculate fluxes interactively based on the observed SST of 294.9 K, we choose to impose surface fluxes taken from the VARANAL product. The fluxes are time-varying but have mean values of 11.8 W m$^{-2}$ (sensible) and 105.8 W m$^{-2}$ (latent) over the simulation period. The surface

stress is a constant imposed value of 0.0625 m$^2$ s$^{-2}$. SAM configuration files are located in at https://github.com/dmechem/ENA_variability_LES_bulk_paper.

## 4    Results

### 4.1    LES base state and mean turbulent fluxes

The mean control-simulation profiles in Fig. 4, taken over the averaging period of 0900–1200 UTC, show that stratocumulus

clouds dominate the domain. We define cloud as points having liquid cloud water mixing ratio ($q_c$) of 0.01 g kg$^{-1}$ or greater. The main stratocumulus cloud layer extends from an average cloud-base height of 821 m to an average cloud top height of 1109 m (Fig. 4a). This cloud boundaries agree well with the in situ and ground measurements in Z21 (see their Figure 1 and Figure 4). The PDF of cloud-base heights in Fig. 4e shows the prevalent stratocumulus cloud base and a secondary peak corresponding to shallow cumulus, most of which rise into the stratocumulus deck. Mean cloud base and cloud top are denoted

by the lower and upper gray lines. Cloud fraction is 0.5 at these levels (Fig. 4d), with a fully cloudy domain occurring between 900 m and 1050 m. The liquid water and cloud fraction profiles (Fig. 4a and d) indicate the presence of cumulus occurring below the main stratocumulus deck. These cumulus extend down to 450 m and are characterized by a lower area fraction ($\sim$0.1), therefore contributing only a small amount to the mean $q_c$ profile. The stratocumulus deck is characterized by a nearly linear increase in $q_c$ from the mean cloud base upward, peaking at $\sim$0.5 g kg$^{-1}$ close to the mean cloud top. Liquid water





variance increases only slowly with height from the mean cloud base up to a height of approximately 1.04 km, above which it
rapidly increases, reflective of the effects of entrainment. Variables $q_t$ and $\theta_l$ are weakly stratified in the subcloud layer but are
relatively constant (10.5 g kg$^{-1}$ and 292 K) from 700 m upward to within a few tens of meters of the top of the cloud (Fig. 4b
and c). Variance of $q_t$ and $\theta_l$ is small over the bulk of the boundary layer, only increasing from entrainment near the upper part
of the cloud similar to the $q_c$ variance.

Buoyancy flux slowly increases in the stratocumulus layer, peaking near 20 W m$^{-2}$ in the upper half of the cloud, as shown
in Figure 5a. Entrainment at cloud top is manifested by negative values of buoyancy flux ($w'\theta_v' < 0$, i.e., entrainment acting to
bring warm air downward). Below the main stratocumulus layer, the buoyancy flux shows signs of decoupling in the sub-cloud
layer. This arises due to the evaporative cooling from rain that creates a stable layer below the stratocumulus, requiring the
turbulence to do work against the stable stratification in an attempt to homogenize the layer. This stratification also helps to
explain some of the rising cumulus below the main stratocumulus deck, as the decoupling leads to a buildup of conditional
instability in the mixed layer to support cumulus growth.

Figure 6 shows a snapshot plan view of of the liquid water path (LWP) over the domain taken at 1030 UTC. The LWP is

We use the buoyancy flux to diagnose the depth of the entrainment zone. Although we recognize that the effects of cloud-top
entrainment are undeniably, over time, communicated throughout the boundary layer, we define the entrainment zone to be
the depth to which the entraining free troposphere initially penetrates. We define this as the depth at which the profile of the
buoyancy flux switches direction (i.e., where the derivative of the buoyancy flux is zero). Fig. 5a shows that the gray line at
1.005 km denoting the bottom of the entrainment zone (also overlaid on all the panels of Figs. 4 and 5) coincides with the
maximum of the buoyancy flux and the zero of the derivative of the buoyancy flux. At this level, the variances of $q_c$, $q_t$, and $\theta_l$
begin to increase substantially.

Vertical velocity variance (Fig. 5b) begins to increase in the cumulus layer and peaks in the upper levels of the stratocumulus
layer, consistent with increasing buoyancy production. Positive vertical velocity skewness (Fig. 5c) occurs in the cumulus re-
gion in a manner consistent with other studies showing cumulus (narrow, strong updrafts, and broad, weaker downdrafts) rising
into stratocumulus (Stevens et al., 2001). Skewness over the bulk of the stratocumulus cloud is predominantly negative (narrow,
strong downdrafts and broad, weak updrafts), which is consistent with observations of radiatively driven stratocumulus (e.g.,
Stevens et al., 2003). The increase of skewness near cloud top is associated with entrainment and is ubiquitous in stratocumulus
simulations.

Figure 6 shows a snapshot plan view of of the liquid water path (LWP) over the domain taken at 1030 UTC. The LWP is
highly variable, with maxima over 300 g m$^{-2}$ and minima below 25 g m$^{-2}$. Two arbitrarily selected vertical cross sections
of cloud and rain water mixing ratio show that LWP variability often corresponds to variability in cloud depth. Cloud-base
height is much more variable than cloud-top height, which is typical of stratocumulus whose tops are constrained by a strong
inversion. The vertical slices also show indications of cumulus rising into stratocumulus (Fig. 6c, between 21.5 and 22.0 km).
Regions of rain water mixing ratio in (b) and (c) are, broadly speaking, confined to regions of cloud with higher LWP and cloud
water mixing ratio.



## 4.2 Vertical profiles of cloud horizontal variability

We quantify the horizontal variability of the $q_c$ using its IRV $\nu_{q_c}$ defined in Eq. (2). As mentioned earlier, the larger the value
of $\nu_{q_c}$ the smaller the horizontal variation of $q_c$ in relation to the mean. The vertical profile of $\nu_{q_c}$ in Fig. 7a increases upward
from the base of the stratocumulus layer (the dashed lines at $z = 821$ m), peaking just above the lower extent of the entrainment
zone near 1020 m and then sharply decreasing up to cloud top. The shape of this profile agrees well with aircraft observations
from Zhang et al. (2021) and overplotted on Fig. 7a, which also exhibit an increase in $\nu_{q_c}$ throughout the cloud layer and then
decrease closer to cloud top (see also Fig. 1b).

The portion of the profiles between the mean cumulus and stratocumulus cloud bases (i.e., between 475 and 821 m) exhibits
a decrease in $\nu_{q_c}$ corresponding to an increase in relative variability. The mean $q_c$ throughout the layer is increasing, a con-
sequence the increase of $q_c$ with height from condensation, just as in stratocumulus. The increase in standard deviation with
height is likely a consequence of the cloud-top distribution of the cumulus, specifically, that not all the cumulus rise completely
into the stratocumulus deck. Keeping in mind that these calculations are conditioned on cloudy points at each level, this vari-
ability in cloud-top height over the cumulus layer means fewer cloud samples with height, leading to larger values of standard
deviation.

Breaking down $\nu_{q_c}$ into its constituent mean and standard deviation values (Fig. 7a) demonstrates that the increase in $\nu_{q_c}$
throughout the lower 2/3 of the stratocumulus cloud layer is largely not due to changes in horizontal variability, but rather
mainly because of the adiabatic increase of mean $q_c$ (i.e., the numerator term in (2)). The standard deviation of $q_c$ varies little
over this portion of the cloud layer. In the entrainment zone, the variability of $q_c$ begins to increase with height, while the
mean $q_c$ increases more slowly with height. The behavior of the $q_c$ mean and variability are both likely effects of cloud-top
entrainment, which, combined together, yields the sharp decrease in $\nu_{q_c}$.

To achieve a quantitative understanding of the relative roles of the mean and standard deviation in determining how $\nu_{q_c}$
changes with height, we use the chain rule to take the derivative of $\nu_{q_c}$ with respect to height $z$:

$$\frac{d\nu_{q_c}}{dz} = \frac{d}{dz}\left(\frac{\langle q_c\rangle^2}{\mathrm{Var}(q_c)}\right)$$

$$= \underbrace{\frac{2\langle q_c\rangle}{\mathrm{Var}(q_c)}\frac{d\langle q_c\rangle}{dz}}_{\text{Term 1}} - \underbrace{\frac{\langle q_c\rangle^2}{\mathrm{Var}^2(q_c)}\frac{d\mathrm{Var}(q_c)}{dz}}_{\text{Term 2}} \tag{3}$$

where the first term on the right hand side (term 1) reflects the impact of the vertical variation of mean $q_c$ and the second
term (term 2) reflects the impact of the vertical variation of variance of $q_c$. In the cumulus layer, changes with height of both
mean and variance of $q_c$ contribute to the variation with height of of $\nu_{q_c}$, as shown in Figure 7b. Term 1 is greater near mean
cumulus cloud base, contributing to a positive $d\nu_{q_c}/dz$. Above this maximum, a negative maximum in term 2 is associated
with $d\nu_{q_c}/dz$ becoming negative. In the lower part of the stratocumulus layer, an increasingly positive term 1 dominates the
change in variability of $\nu_{q_c}$, while term 2 (and thus the impact of the variance of $q_c$) is small and then becomes more negative
approaching the base of the entrainment zone. At this point, term 1 remains positive but begins to decrease as adiabatic mean
$q_c$ growth slows, but term 2 becomes increasingly negative due to increased $q_c$ variance variability from the effects of cloud-top





entrainment. Beginning about 20 m above the base of the entrainment zone, $d\nu_{q_c}/dz$ becomes negative up to cloud top as term 2 remains the dominant term from entrainment and term 1 (which is decreasing with height) becomes less important.

The above analysis clearly demonstrated the advantages and usefulness of LES in comparison with in situ observations. First of all, the thermodynamic structure of MBL and stratocumulus cloud from the LES shed important light on the underlying physical processes (i.e., condensation and entrainment) that influence the cloud vertical structure and horizontal variation.

Second, the LES better resolves the vertical structure of the cloud field, which allows us to perform the decomposition analysis as in Fig. 7 to obtain a more confident and comprehensive understanding of the vertical dependence of $q_c$ horizontal variability.

### 4.3    Vertical profiles of enhancement factor

In the single-moment microphysical parameterization used in these simulations, the enhancement factor $E_q$ can be formulated as in Zhang et al. (2019) to be:

$$E_q = \frac{\int_0^\infty q_c^{\beta_q} P(q_c) dq_c}{\langle q_c \rangle^{\beta_q}} \tag{4}$$

where $P(q_c)$ represents the probability density function of $q_c$. Because we hold $N_c$ constant in this study, we consider only the variability in cloud water in formulating $E_q$. In the cumulus region, $E_q$ varies between 1 and 2 and slowly increases with height approaching the mean stratocumulus base, peaking at 3.80 near 800 m (Fig. 8a). Above cloud base, $E_q$ sharply decreases to approximately 1 in the upper 1/3 of the stratocumulus layer. In the entrainment zone, $E_q$ begins to slightly increase

and approaches 2 above mean cloud top. Model calculations of $E_q$ are similar to those calculated from aircraft observations from the case (Zhang et al., 2021), both over the stratocumulus layer itself and just below the mean stratocumulus cloud base. Aircraft observations overlaid on Fig. 8a correlate well with calculated $E_q$, agreeing with the idea of a decreasing $E_q$ profile throughout the cloud layer. The previous analysis of $\nu_{q_c}$ suggests that the large vertical variations of $E_q$ in the stratocumulus layer are predominantly due to adiabatic increase of mean $q_c$, whereas the impact of $q_c$ variances are more minimal until closer

to cloud top.

A lognormal distribution has been shown to represent observations of $q_c$ well (e.g., Lebsock et al., 2013; Zhang et al., 2019), so we represent $E_q$ in terms of the parameters of a lognormal distribution as in (Zhang et al., 2021):

$$E_q(\nu_{q_c}, \beta_{q_c}) = \left(1 + \frac{1}{\nu_{q_c}}\right)^{\frac{\beta_{q_c}^2 - \beta_{q_c}}{2}} \tag{5}$$

where $E_q$ represents the enhancement factor related solely to variability in $q_c$. Equation 5 nicely demonstrates the dependence

of $E_q$ on the inverse relative variance. In particular, (5) shows that a PDF of $q_c$ undergoing adiabatic ascent and shifting toward larger values of $q_c$ but maintaining its variance will exhibit a larger mean values of $q_c$, thus causing an increase of $\nu_{q_c}$ and a reduction of $E_{q_c}$.

Using this definition of of $E_q$ based on the lognormal assumption of the $q_c$ distribution, we can differentiate it with respect to height to explore the association between vertical gradients in $E_q$ and vertical profiles of mean and standard deviation of $q_c$:





$$\frac{dE_q}{dz} = \frac{d}{dz}\left(1+\frac{1}{\nu_{q_c}}\right)^{\gamma}$$

$$= \gamma\left(1+\frac{1}{\nu}\right)^{\gamma-1}\left(\frac{-1}{\nu_{q_c}^2}\right)\frac{d\nu_{q_c}}{dz} \tag{6}$$

where $\gamma = (\beta_{q_c}^2 - \beta_{q_c})/2$. Using the expression for $d\nu_{qc}/dz$ from (3), we obtain

$$\frac{dE_q}{dz} = -\frac{\gamma}{\nu_{q_c}^2}\left(1+\frac{1}{\nu}\right)^{\gamma-1}\left(\frac{2\langle q_c\rangle}{\mathrm{Var}(q_c)}\frac{\partial\langle q_c\rangle}{\partial z} - \frac{\langle q_c\rangle^2}{\mathrm{Var}^2(q_c)}\frac{\partial\mathrm{Var}(q_c)}{\partial z}\right)$$

$$= \underbrace{-C\frac{2\langle q_c\rangle}{\mathrm{Var}(q_c)}\frac{\partial\langle q_c\rangle}{\partial z}}_{\text{Term 1}} + \underbrace{C\frac{\langle q_c\rangle^2}{\mathrm{Var}^2(q_c)}\frac{\partial\mathrm{Var}(q_c)}{\partial z}}_{\text{Term 2}} \tag{7}$$

for the decomposition of the vertical gradient of $E_q$. In (7), $C$ is defined as follows, without the negative sign:

$$C = \frac{\gamma}{\nu_{q_c}^2}\left(1+\frac{1}{\nu}\right)^{\gamma-1}$$

This allows us to utilize similar analysis used in the gradient of $\nu_{q_c}$ to break down components into impacts from mean $q_c$ and variance of $q_c$, as shown in Figure 8. The constant is defined such that an increase in variance with height corresponds to an increase in $E_q$ with height, and an increase in mean $q_c$ with height corresponds to a decrease in $E_q$ with height.

This estimate of $dE_q/dz$, calculated assuming a lognormal distribution of $q_c$, matches fairly well to the actual derivative of $E_q$ calculated from the unapproximated PDF (Fig. 8b). Above the cumulus cloud base ($\sim$475 m), $dE_q/dz$ slowly increases as term 2 remains almost constant but term 1 approaches zero. Approaching the stratocumulus cloud base at 750 m, $dE_q/dz$ spikes to near 0.03 m$^{-1}$ as term 1 increases but term 2 decreases to zero. $dE_q/dz$ then sharply decreases and becomes negative until above the main stratocumulus cloud base, close to 850 m. This is driven by a sharp decrease in term 1 while term 2 is also negative but increasing to near zero. Throughout most of the stratocumulus cloud layer, the vertical gradient of $E_q$ is negative due to a negative term 1 and an almost constantly zero term 2 but slowly increases to zero. At the base of the entrainment zone, $dE_q/dz$ becomes zero and then slightly positive as term 2 becomes more dominant, owing to increased variability of $q_c$ due to entrainment.

### 4.4   Dependence on droplet concentration

Our simulations use single-moment microphysics where the $N_c$ is specified but is constant throughout the simulation. Smaller droplet concentrations tend to promote precipitation, which should lead to an increase in variability of cloud and precipitation variables. To explore the sensitivity of $N_c$ on inverse relative variance and enhancement factor, we performed simulations over a range of $N_c$ values. As previously described, these runs employ values of 50 and 100 cm$^{-1}$ to compare to our control simulation of 75 cm$^{-1}$. Mean base state profiles in Fig. 9 show predictable trends across all three sensitivity simulations. Each
domain has cloud fraction of about 0.1 in the cumulus region and a fully cloudy domain in the stratocumulus layer (Fig. 9d). Cloud bases and cloud heights vary, though, with mean cloud bases increasing by $\sim$10 m but mean cloud tops increasing by





closer to 25 m with each increase in $N_c$. The decrease in cloud-top height with decreasing $N_c$ likely corresponds to a decrease in entrainment associated weaker turbulence accompanying the stabilizing effects of increased precipitation (e.g., Stevens et al., 1998). Mean $q_c$ profiles increase similarly, with the deeper clouds associated with larger $N_c$ having greater $q_c$ (Fig. 9a).

$q_t$ and $\theta_l$ profiles are virtually identical from the surface to mean stratocumulus cloud top, where the profiles follow the same trend but again are shifted upward in height by about 30 m with each increase in $N_c$ (Fig. 9b and c). Above the cloud layer (∼1200 m), these profiles become identical once again. Variances of $q_c$, $q_t$, are much closer in value to one another but each has the same 30 m height increase associated with increasing $N_c$.

Although main stratocumulus cloud bases differ slightly, the probability density function (PDF) of cloud base height indi-
cates that a stratocumulus cloud base height of ∼825 m is most common for each $N_c$ sensitivity simulation. The maximum of the PDF is nearly identical for $N_c$ values of 100 and 75 cm$^{-1}$ at 0.36 (Fig. 9e). The 50 cm$^{-1}$ simulation peaks near 825m as well, but it is more of a plateau between 800m and 825m with a lower probability of 0.26. The PDF also shows a second, smaller peak in cloud base height the cumulus region. The cumulus region maximum PDF values are about 0.08, but the corresponding cloud base values differ slightly across the simulations. The peak frequency occurs near 525 m for a $N_c$ value of
100 cm$^{-1}$, but it is closer to 475 m for the other $N_c$ values.

Further analyzing the impact of the $N_c$ on sub-grid scale variability, we calculate the inverse variance $\nu_{q_c}$ and enhancement factor $E_q$ for each $N_c$ simulation. $\nu_{q_c}$ is consistent across the different $N_c$ runs with similar features occurring in both the cumulus and stratocumulus cloud layers (Fig. 10a). $\nu_{q_c}$ corresponding to $N_c$ of 100 cm$^{-1}$ is the largest among the three simulations. In the stratocumulus layer, the smallest value of $N_c$ (50 cm$^{-1}$) corresponds to the smallest value of $\nu_{q_c}$, likely
a result of the stabilizing effects of precipitation and the associated reduction in vertical moisture flux from low levels. $E_q$ values for the 3 $N_c$ are even more similar to one another than for $\nu_{q_c}$, with some differences near the stratocumulus cloud base (Fig. 10b).

Although not directly accounting for the variability of $N_c$, this sensitivity study demonstrates that even with varying fixed $N_c$ values, results show consistent trends among key variables. The overall cloud structure and base states are almost identical,
differing slightly in magnitude and shifted in height. An analysis of $\nu_{q_c}$ demonstrates similar increasing trends throughout the stratocumulus layer like in the large domain control simulation. More importantly $E_q$ shows a clear, almost identical decrease over the stratocumulus layer between cloud base and roughly the beginning of the entrainment zone. Although the differences in $\nu_{q_c}$ over the upper part of the cloud are substantial across the simulations, the large values of $\nu_{q_c}$ translate only to small differences in small values of $E_q$. Because of this, the choice of a constant $N_c$ value does not greatly impact the value of $E_q$.

## 5 Discussion and Conclusions

One of the major uncertainties in warm rain simulations within ESMs is accounting for microphysical processes occurring at the subgrid scale. Jensen's inequality indicates that neglect of subgrid-scale variability may give rise to biases in nonlinear process rates such as autoconversion. Some current ESMs employ a multiplicative enhancement factor to process rates to





crudely account for sub-grid scale variability. In this study, we use LES to explore the behavior of the quantities determining
the autoconversion enhancement factor. Our chief findings are as follows:

- Both inverse relative variance $\nu_{q_c}$ and enhancement factor $E_q$ vary considerably in the vertical.

- The profile of inverse relative variance $\nu_{q_c}$ throughout most of the stratocumulus and underlying cumulus layers is explained largely by changes to the mean $q_c$ profile, predominantly dictated by adiabatic processes and not by variability in $q_c$.

- The profile of enhancement factor $E_q$ is largely consistent with that of $\nu_{q_c}$ and peaks near cloud base, with a minimum over the upper half of the cloud. $E_q$ is governed by the inverse relative variance and not simply the absolute width of the distribution, so a $q_c$ distribution undergoing only adiabatic ascent will yield a reduction of $E_q$.

- Over the upper 1/3 of the stratocumulus layer where entrainment is highly influential, the increased variance of $q_c$ has a substantial effect on $\nu_{q_c}$ and $E_q$.

- Because $E_q$ is so strongly governed by adiabatic (condensational) increase of $q_c$ with height, which is a structural feature of MBL stratocumulus, our results likely generalize well beyond this single case.

- Representing the $q_c$ distribution using a lognormal function is a reasonable approximation.

The strong vertical variation in $E_q$ suggests that a constant, global value of $E_q$ is an oversimplification, at least for a typical case of marine stratocumulus like we are exploring here. Moreover, a constant value of 3.2, applied everywhere in the cloud, is likely too large. In the middle and upper part of the cloud where autoconversion is most likely occurring, $E_q$ has a smaller value between 1 and 1.5. Only near cloud base does $E_q$ attain large values over 3.0, yet these regions have relatively little liquid water and therefore likely exhibit very little autoconversion.

This analysis considers only variability in cloud water $q_c$. A 2-moment or size-resolving (bin) microphysical parameterization would take into account variability in $N_c$, a dependence that is also nonlinear (e.g., $N_c^{-1.79}$ in Khairoutdinov and Kogan (2000)). This additional nonlinear term provides an additional pathway for subgrid-scale variability bias, although recent work has demonstrated that the covariance between $q_c$ and $N_c$ largely counteract the individual effects of $q_c$ and $N_c$ variability (Zhang et al., 2019, 2021). Extending this work using bin microphysics is the object of a future study, which will allow us to consider not only the nonlinearity associated with $N_c$ but also the effect on enhancement factor of homogeneous vs. inhomogeneous mixing regimes (at least down to the limited spatial and temporal scales associated with the model grid and time step) in the entrainment zone.

*Author contributions.* JAC and DBM performed LES analysis and created most of the figures. ZZ provided in-situ observations and drafted the case study section and the associated figure. JAC, DBM, and ZZ contributed to experimental design and writing of the manuscript.



*Competing interests.* The authors declare that they have no conflict of interest.

*Acknowledgements.* J. Covert and D. Mechem was supported by subcontract OFED0010-01 from the University of Maryland Baltimore
410  County and the U.S. Department of Energy's Atmospheric Systems Research grant DE-SC0016522. Co-author Z. Zhang was supported by
Atmospheric System Research Grant DE-SC0020057 funded by the Office of Biological and Environmental Research in the US DOE Office
of Science.



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

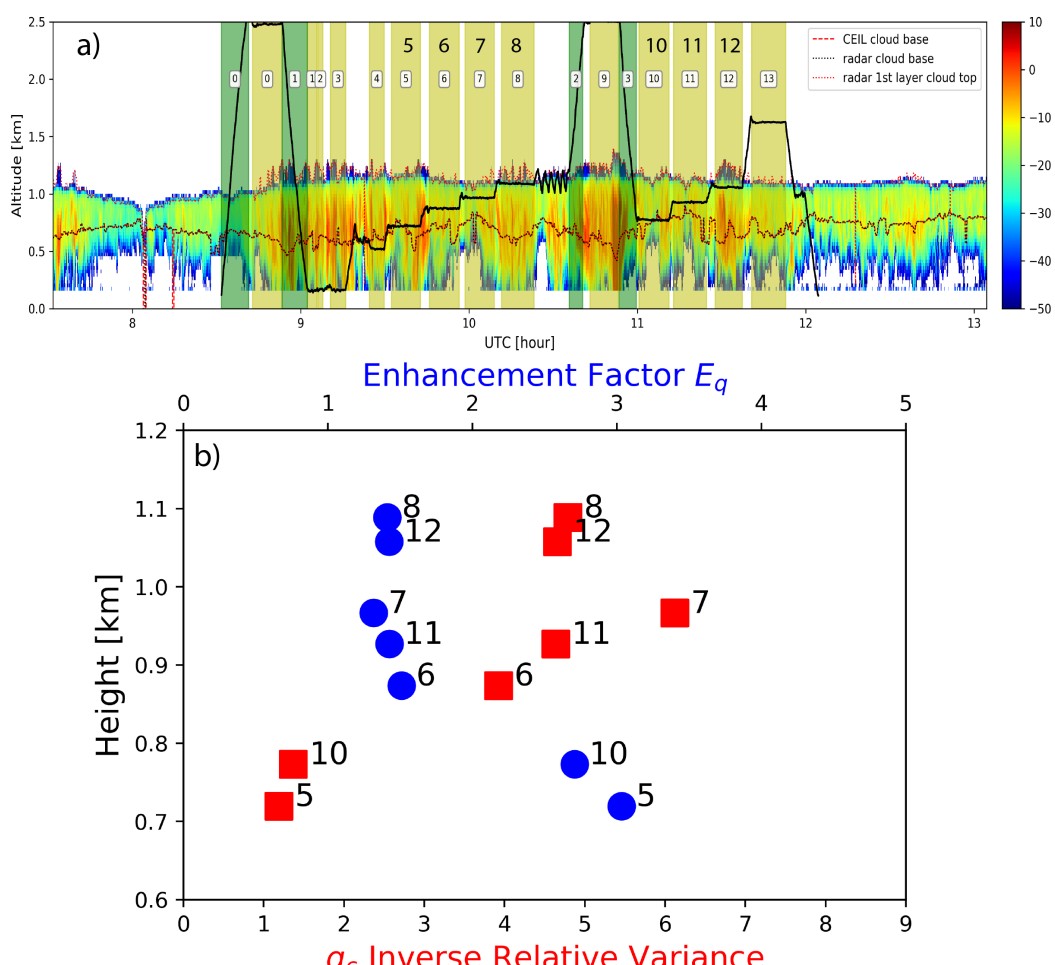

**Figure 1.** (a) The vertical flight track of G-1 (thick black line) overlaid on the radar reflectivity contour by the ground-based KZAR during the 18 July 2017 RF around the DOE ENA site on Graciosa Island. The dotted lines in the figure indicate the cloud base and top retrievals from ground-based radar and ceilometer instruments. The yellow-shaded regions are the "hlegs" identified by the Z21 study, among which seven are selected. See text for their definitions. (b) IRV $\nu_{q_c}$ and enhancement factor $E$ as a function of height that are derived from the selected hlegs 5, 6, 7, 8, 10, 11, and 12 in (a).

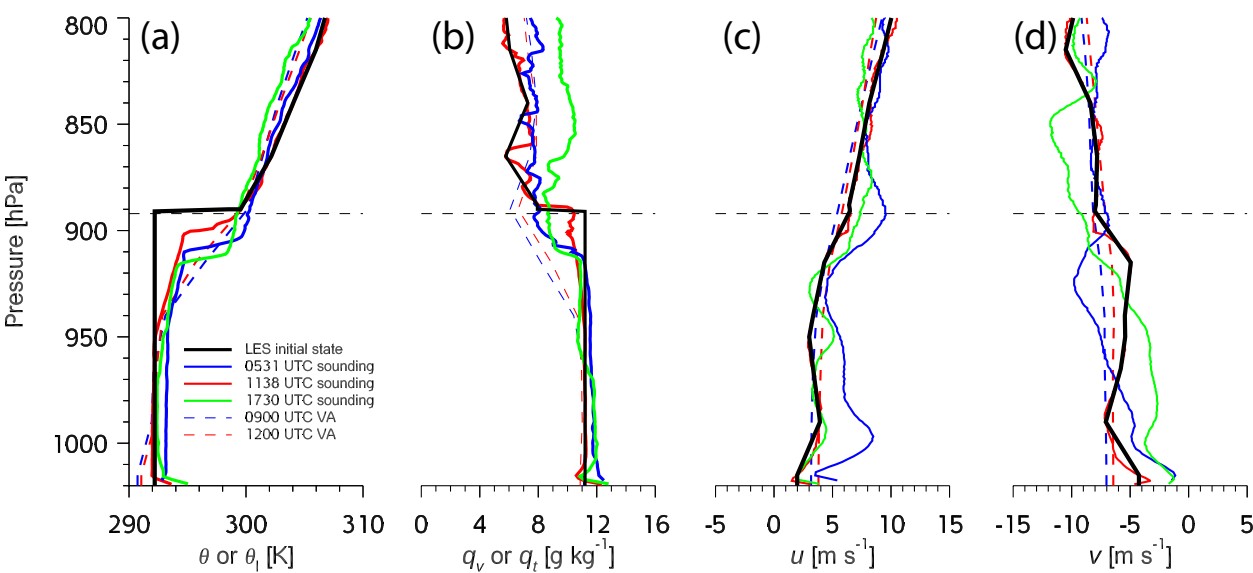

**Figure 2.** Vertical profiles of the 18 July 2017 boundary layer from radiosondes taken at Graciosa Island (0581, 1138, and 1730 UTC) and the ARM VARANAL product (0900 and 1200 UTC), which itself assimilates the soundings. (a) Potential temperature or liquid water potential temperature (LES profile). (b) Water vapor mixing ratio or total water mixing ratio (LES profile). (c) $u$. (d) $v$. The solid black line represents the simplified LES initial condition profile constructed from the 1138 UTC sounding.



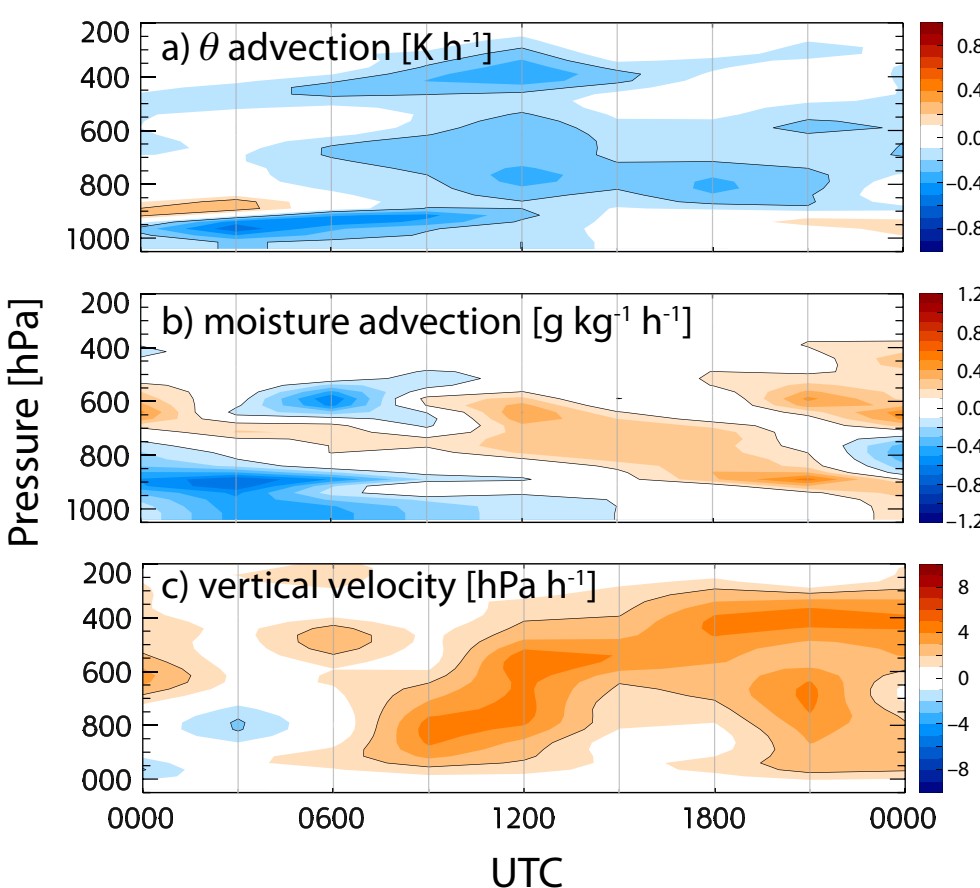

**Figure 3.** Time-height sections of potential temperature and moisture advection, and large-scale vertical velocity taken from the ARM VARANAL product. The simulation period is from 0600–1500 UTC.

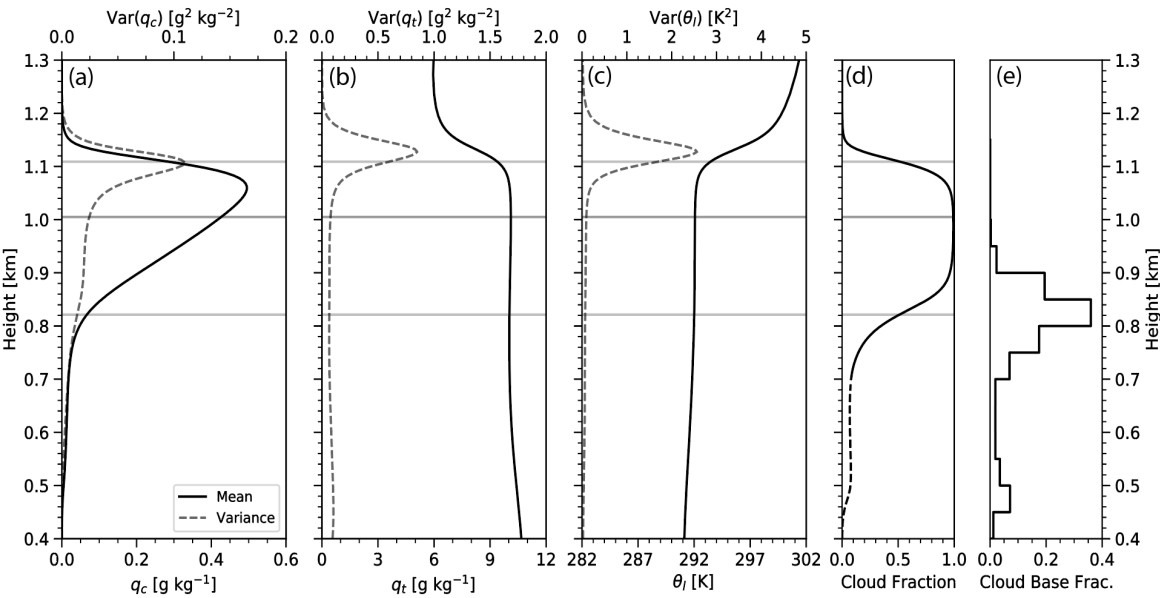

**Figure 4.** Vertical profile of horizontally averaged (mean) and variance over the analysis period of 0900–1200 UTC of (a) cloud liquid water mixing ratio $q_c$, (b) total water $q_t$, (c) liquid water potential temperature $\theta_l$, and (d) cloud area fraction. The solid lines indicate mean quantities and the dashed lines indicate variance. From bottom to top, the gray lines indicate mean cloud base, bottom boundary of the entrainment zone, and mean cloud top. (e) histogram of cloud base height, with each value representing the area fraction of cloud bases lying within that 50-m height interval.

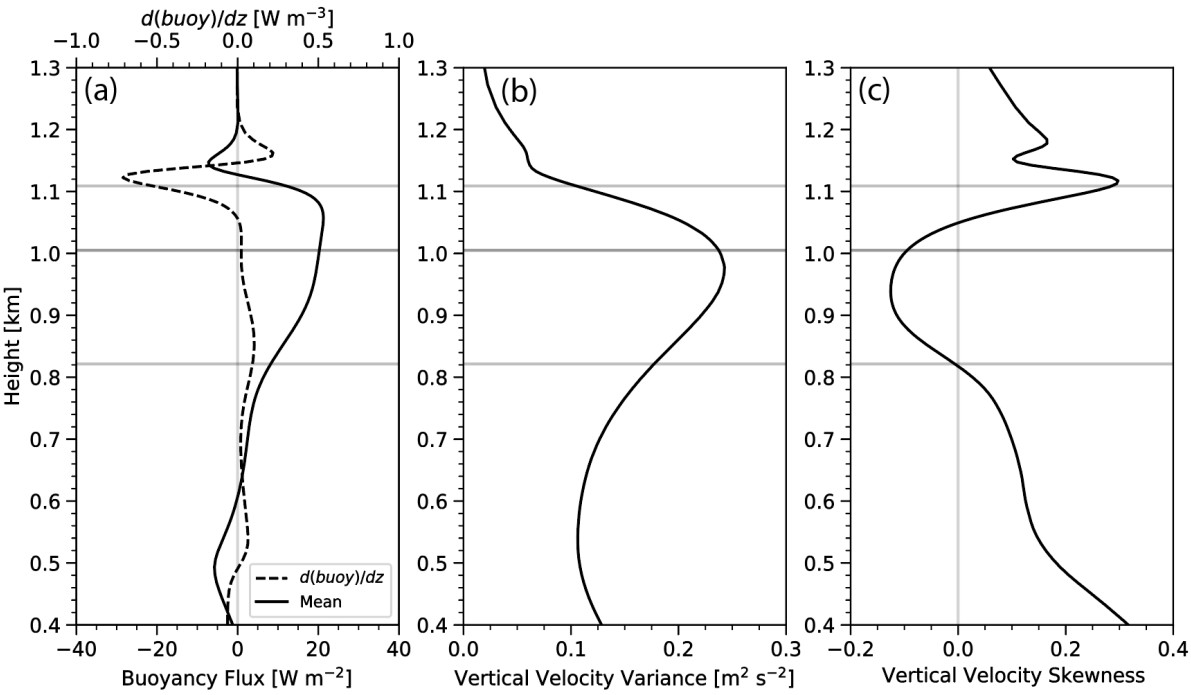

**Figure 5.** Vertical profiles of horizontally averaged (a) buoyancy flux, (b) vertical velocity variance, and (c) vertical velocity skewness. The dashed line in (a) represents the first derivative of the buoyancy flux. The gray lines are as in Fig. 4.



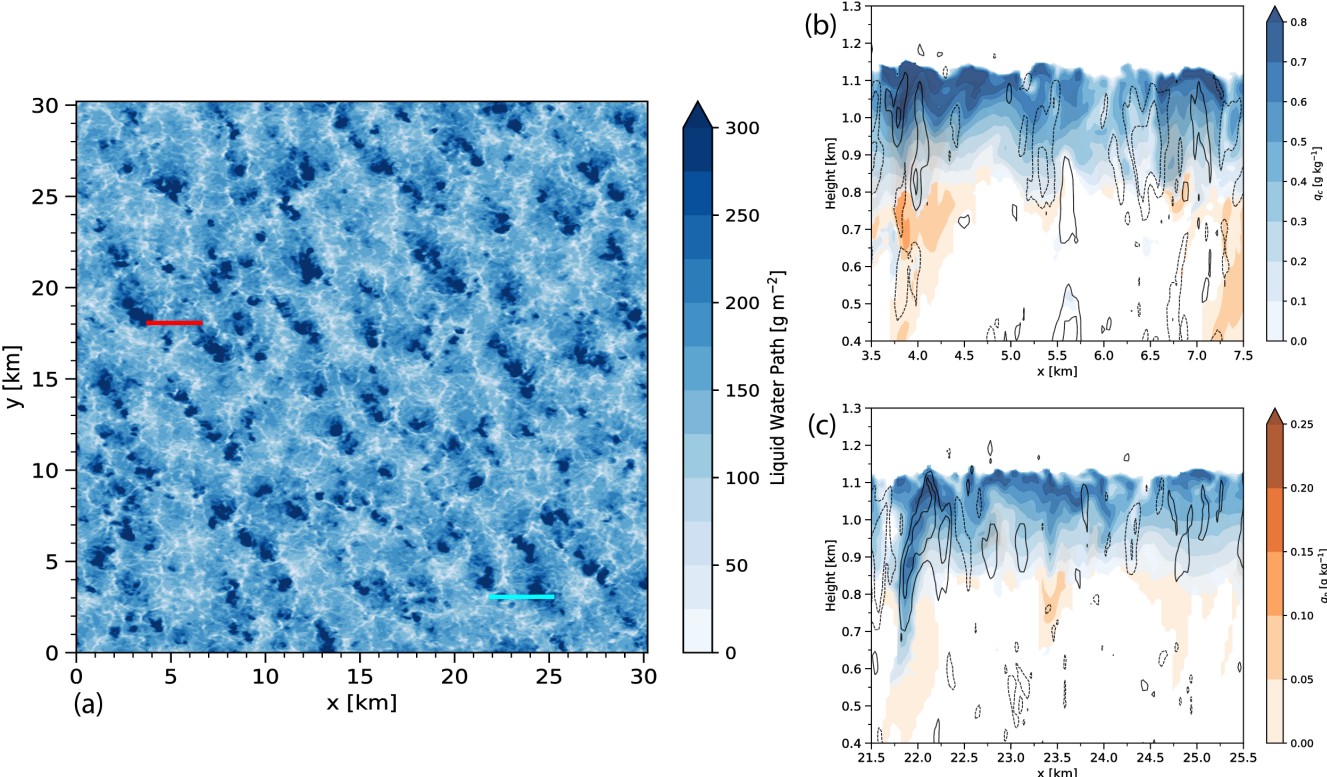

**Figure 6.** (a) Plan view of (a) cloud liquid water path. (b) and (c) correspond to vertical cross sections of cloud water mixing ratio ($q_c$, blue), rain water mixing ratio ($q_r$, orange), and contours of vertical velocity $w$. $w$ is contoured every 0.5 m s$^{-1}$ with solid lines representing positive values and dashed lines representing negative values. The red and blue lines on (a) correspond to the locations of the vertical cross sections in (b) and (c), respectively.



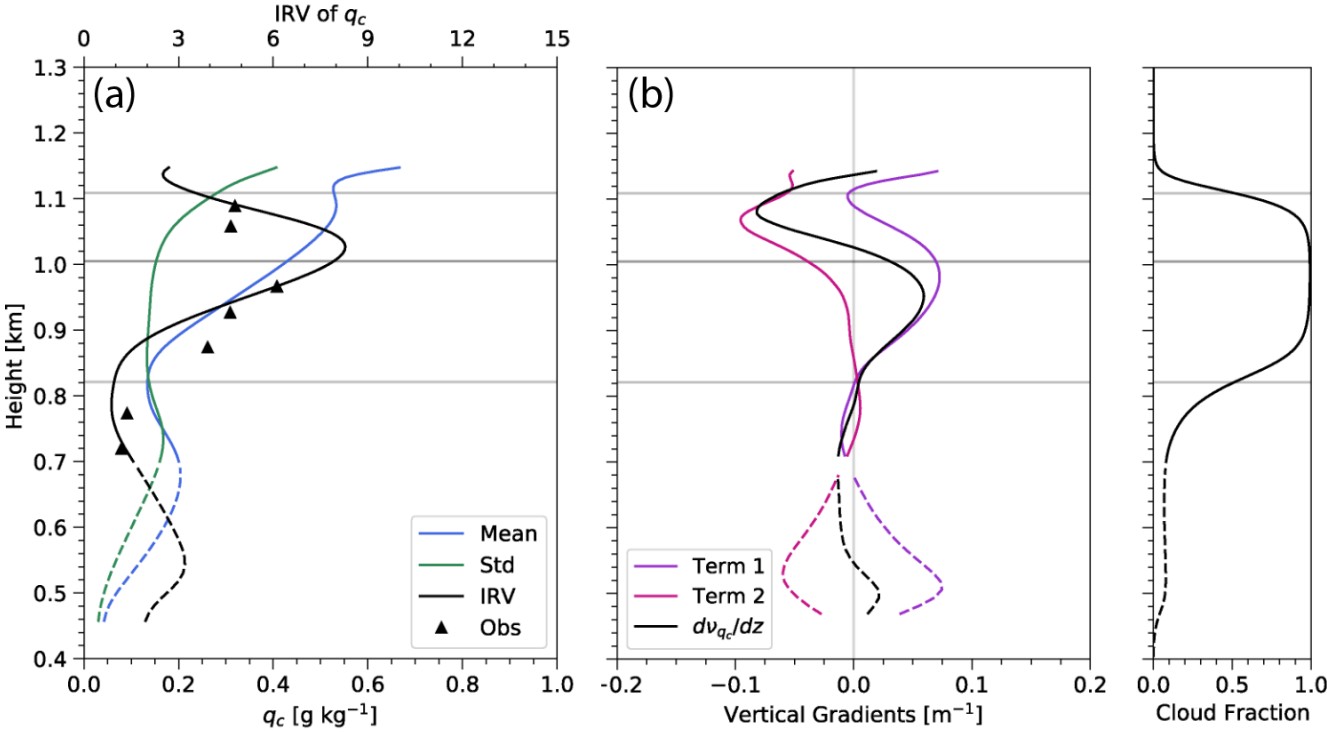

**Figure 7.** Vertical profiles of horizontally averaged (a) mean, standard deviation, and inverse variance of $q_c$ and (b) derivatives of the inverse variance of $q_c$. Dashed lines represent the cumulus cloud layer. Term 1 and term 2 are calculated as defined in Eq. 3. The gray lines are as in Fig. 3.



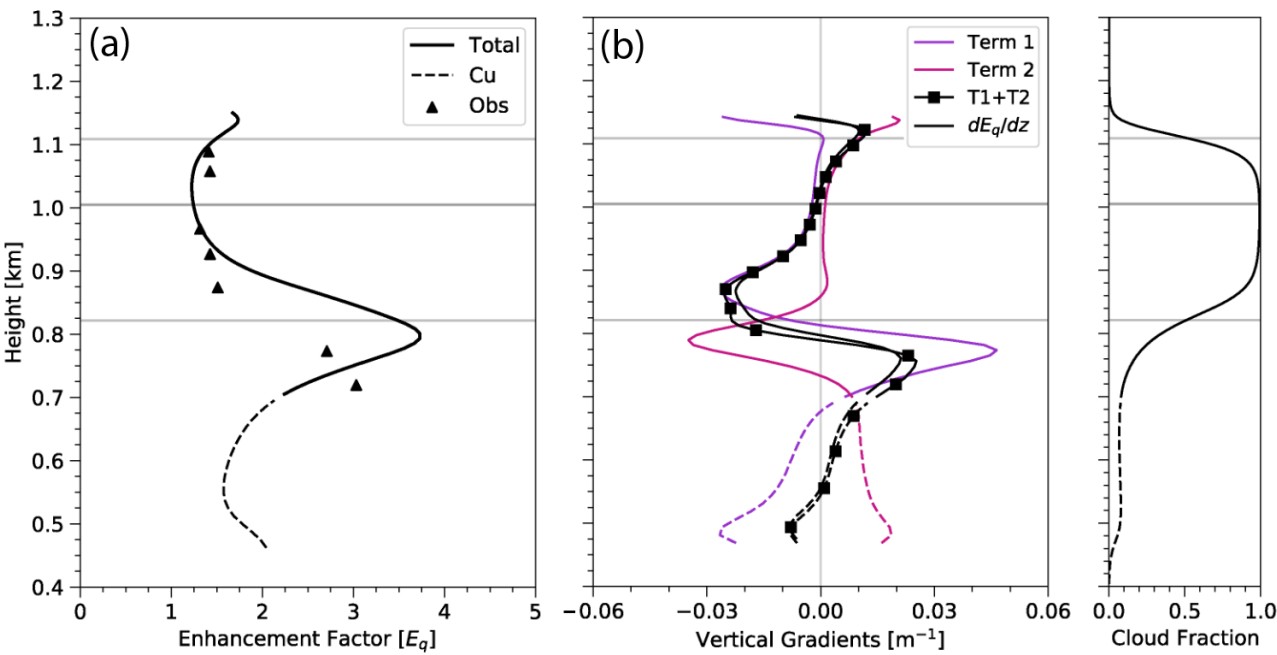

**Figure 8.** Vertical profiles of horizontally averaged (a) enhancement factor $E_q$ and (b) derivatives of the enhancement factor $E_q$. As in Fig. 7, dashed lines represent the cumulus cloud layer. Term 1 and term 2 are calculated as defined in Eq. 3. The gray lines are as in Fig. 4.

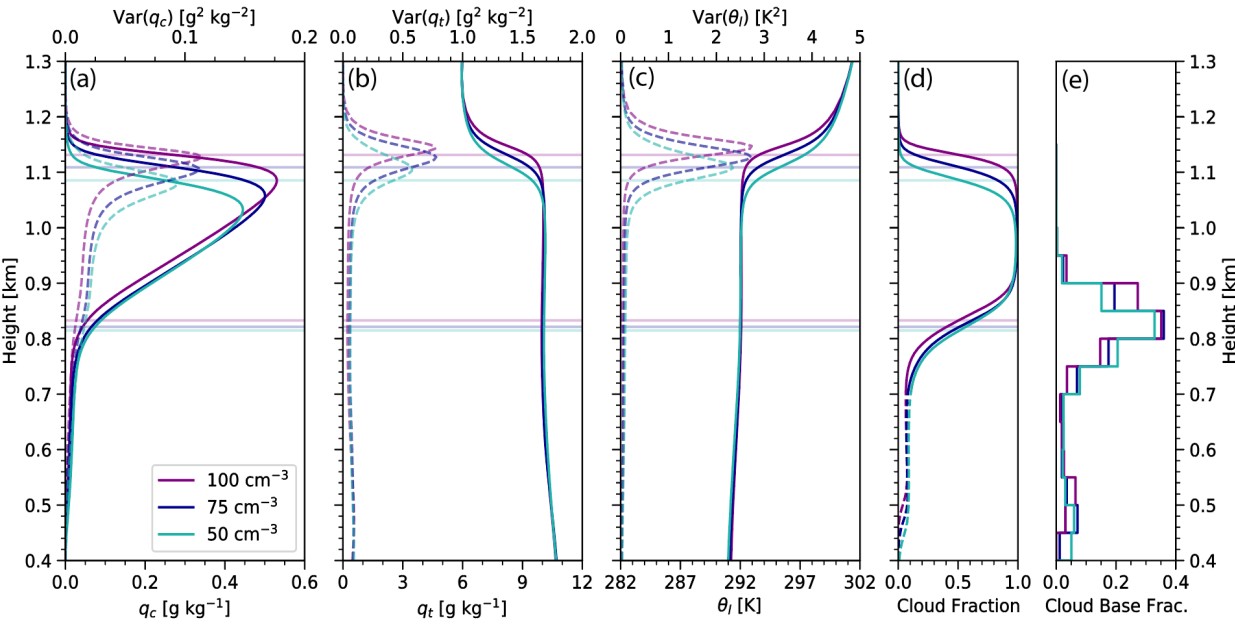

**Figure 9.** Vertical profile of horizontally averaged (mean) and variance, averaged over the analysis period of 0900–1200 UTC. (a) cloud liquid water mixing ratio $q_c$. (b) total water $q_t$. (c) liquid water potential temperature $\theta_l$. (d) Cloud area fraction. The solid lines indicate mean quantities and the dashed lines indicate variance. (e) histogram of cloud base height, with each value representing the area fraction of cloud bases lying within that 50-m height interval. Purple, dark blue, and teal lines represent simulations with $N_c$ values with 100, 75, and 50 cm$^{-3}$, respectively. The colored lines correspond to cloud base and height as in Fig. 4.

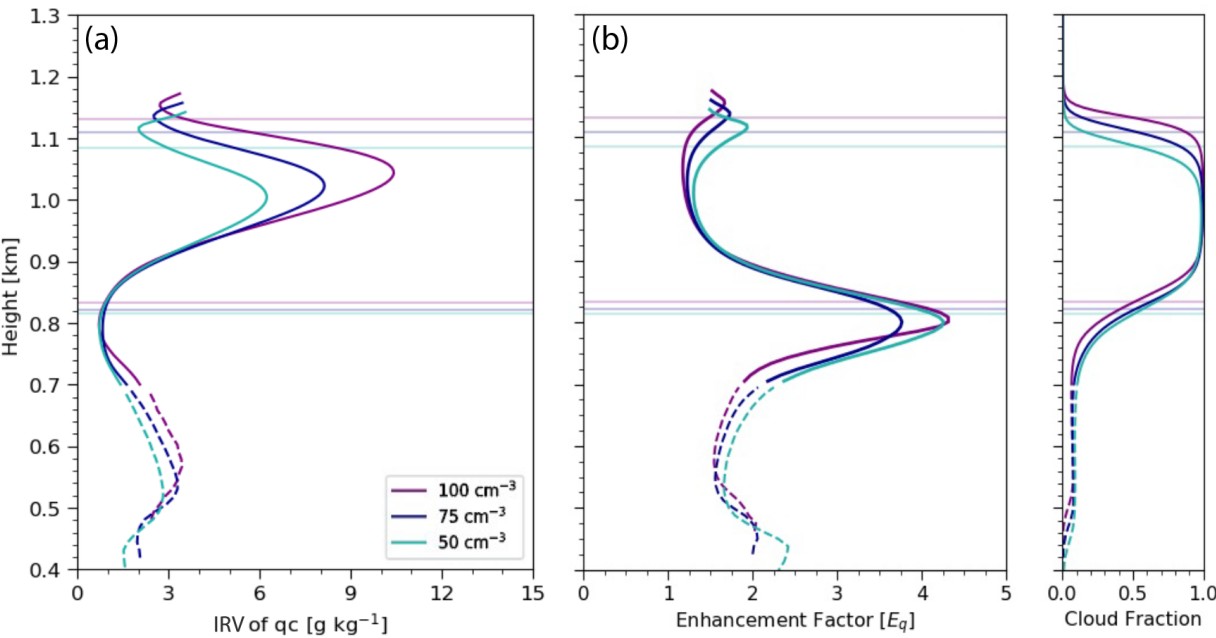

**Figure 10.** Vertical profiles of horizontally averaged (a) inverse variance $\nu_{q_c}$ and (b) enhancement factor $E_q$. Solid lines represent total values while dashed lines represent values correlating to only cumulus clouds. Purple, dark blue, and teal lines represent simulations with $N_c$ values with 100, 75, and 50 cm$^{-3}$, respectively. The colored lines correspond to cloud base and height as in Fig. 4.