# Peer review of "Subgrid-scale Horizontal and Vertical Variations of Cloud Water in Stratocumulus Clouds: A case study based on LES and comparisons with in-situ observations"

_Atmospheric Chemistry and Physics, 2021_

## Author Response (AR1)

**Response to Reviewers**

"Subgrid-scale Horizontal and Vertical Variations of Cloud Water in Stratocumulus Clouds: A case study based on LES and comparisons with in-situ observations"
By Justin A. Covert, David B. Mechem, and Zhibo Zhang
Atmospheric Chemistry and Physics

We greatly appreciate the thoughtful and constructive suggestions from the three reviewers.

Our point-by-point responses follow. Reviewer comments are plain text, and our responses are in blue.

**Response to Reviewer #1**

This relatively straightforward paper uses LES of stratocumulus forced by the VARNAL dataset during ACE-ENA to quantify the vertical variations in the autoconversion enhancement factor (E) and diagnose what causes those variations. The results regarding the vertical variations in E are consistent with observations published previously. The authors find that, in fact, the adiabatic increase in cloud water is the primary source for increasing E at cloud base whereas at cloud top entrainment effects on the cloud water variance have an important effect on E. They conclude that these vertical variations are important to prescribing enhancement factors in low-resolution global models. The data and methods are both appropriate and clearly described. The presentation is of high quality. I have only minor comments listed below.

Fig 1 caption: KZAR -> KAZR.

Thanks to the reviewer for catching this error. We have corrected the text.

Fig 1: You should mention in the caption or add a legend for what red squares and blue dots correspond to.

Thank you to the reviewer for pointing out this lack of clarity in the caption. We have edited the caption as follows:

"...(b) IRV $v_{q_c}$ (red squares) and enhancement factor E (blue circles)…"

Page 5, Line 145: this -> these

Thanks to the reviewer for catching this error. We have corrected the text.

Page 6, line 159: I don't think peaks is the right word here. It is a minimum. 'E is seen to first decrease from cloud base upward until it peaks at around 1 km (i.e., hleg 7), and then increase slightly toward cloud top'

Thanks to the reviewer for catching this error. We have corrected the text as follows:

"... E first decreases from cloud base upward until reaching a minimum around 1 km…"

Page 7, line 197: 'The moisture profile seems to better represent the inversion structure compared with the potential temperature structure, which is more diffuse'. It is not at all clear what is meant by this statement. What is diffuse? How do you judge this representativeness? Also the statement doesn't seem important to your narrative. Either clarify what is meant or just get rid of it.

Thank you to the reviewer for pointing out this confusing statement. We have removed it from the manuscript.

Discussion and Conclusions: Most global models will not resolve the vertical structure of a typical stratocumulus. Can you speculate a bit on the relevance of your results to the relevance of global models that don't resolve the kind of structure you show here.

[Figure]

[Figure caption. Adaptation of Fig. 8a, with an additional line corresponding to values of enhancement factor calculated over layers centered on the E3SM vertical grid points.]

We appreciate the reviewer's comment on the historical challenge ESMs have in trying to resolve thin marine stratocumulus and therefore the difficulty in accounting for the vertical dependence of $E_q$ when only a few model grid points lie inside the cloud. In the figure above (adapted from Fig. 8a), we overlay a profile of $E_q$ calculated over the layers centered on the E3SM model grid points. Older ESMs have historically had coarse vertical grids that may

have had only 1 or 2 points in the cloud, but the figure above shows that even newer ESMs with more grid points (e.g., E3SM, which has 72 vertical points) yields only 4 grid points in the stratocumulus layer, plus a number of additional points corresponding to the underlying cumulus.

With so few cloudy points in the profile, ESMs could use something in the way of a representative constant value. Although the 3.2 value is used in E3SM, both our findings and previous observational studies (e.g., Zhang et al. 2021) indicate that using this single $E_q$ value of 3.2 over the entire cloud is not realistic. We note that $E_q$ values of ~3.2 lie in the lower part of the cloud where autoconversion is very small, and for this reason, finding a single representative value of $E_q$ would require some thought (i.e., should it be an average weighted by autoconversion rate?). Because the figure above is specific to E3SM, we have not included it in the revised manuscript. We have revised the manuscript as follows:

"The strong vertical variation in $E_q$ suggests that a constant, global value of $E_q$ is an oversimplification, at least for a typical case of marine stratocumulus like we are exploring here. While some recently updated ESMs have the vertical resolution to resolve stratocumulus features and include a vertical dependence on $E_q$, many are too coarse and simply cannot. As such, in cases where an ESM must have a constant value of $E_q$, an $E_q$ of 3.2 applied everywhere in the cloud is likely too large and should be reduced. In the middle and upper part of the cloud where autoconversion is most likely occurring, $E_q$ has a smaller value between 1 and 1.5. Only near cloud base does $E_q$ attain large values over 3.0, yet these regions have relatively little liquid water and therefore likely exhibit very little autoconversion."

**Response to Reviewer #2**

SUMMARY
This study analyzes the effects of small-scale variability of cloud liquid water amount on large-scale microphysical process rates using large eddy simulation (LES) of a single, idealized case study over the ARM Eastern North Atlantic site. Simulation results are compared to observational analysis performed over a comparable spatial scale to the LES domain, with generally close agreement found. The study is timely and, while somewhat oversimplified due to the idealized modeling framework and use of the "case study" paradigm, draws some general conclusions about the vertical structure of horizontal atmospheric variability that are of interest to the ACP audience. I have several major concerns that I would like to see the authors address as well as numerous minor and typographical notes. As such, I recommend the manuscript be returned to the authors for major revisions.

MAJOR COMMENTS

The discussion of results and implications is great insofar as it describes the results of a single simulation. But the comparison across simulations with differing domain sizes as well as the explicit suggestion that the results of the study are generalizable are not justified by the arguments presented.

First, concerning the results across domain sizes: the mean profiles are unlikely to change much as a function of domain size as long as the updrafts and downdrafts dictating cloud organization are well-resolved. This is clearly the case with a nearly 9 km horizontal domain. On the other hand, the variances of microphysical fields have been shown to increase with averaging length scale (some references you cite, I also recommend adding Lebo et al. 2014, Wu et al. 2018 and Witte et al. 2019). This implies that comparing the profiles of variances of thermodynamic and microphysical properties across simulations with widely differing domain size is highly ambiguous. While I realize this is not the main point of your paper, this was one of my main conclusions from looking at Figure 10b. The lack of discussion of the noticeably lower maximum Eq factor near cloud base for the $Nc=75$ $cm^{-3}$ case with respect to the smaller-domain simulations leads me to think the authors have not considered the effect of this aspect of model configuration. It is a strong result that LES compares favorably with the observed enhancement factors, both at 30 km. But as a greater variety of grid spacings are used in GCMs and regionally-refined grids become more prevalent, consideration of the scale-dependence of the enhancement factor becomes paramount. This is precisely *because* the variances change but the means don't – the IRV and therefore Eq scale with qc variance. I think the authors can easily address this by running an additional Nc=75 cm-3 simulation on the 8.96 km wide domain. This would both demonstrate the domain-size scaling relationship as well as give appropriate context for analyzing changes in Nc.

Our original manuscript was not clear. All of the profiles originally plotted in the figure actually *were* run on the 8.96x8.96 $km^2$ grid and therefore all the profiles were calculated at the same scale. We clarified this point in the Fig. 10 caption and redrafted the figure to include the profiles derived from the larger-domain control simulation as well.

We agree that scaling is an important factor when dealing with variability. We have cited the papers that you suggest in order to strengthen this point. We agree that as scale increases, variability does as well, to a point. In order to better understand this scaling dependence for our specific case, we ran more simulations over a series of domain sizes and plotted the qc variability as shown in the figure below. Starting at a domain of ~0.5x0.5 $km^2$ (16x16), qc variability gradually increases with domain size but plateaus around our 128x128 run corresponding to a domain of ~4.5x4.5 $km^2$. At domain sizes (scales) bigger than this, qc variability is nearly identical, so the enhancement factor would therefore be very similar, assuming that the mean qc is unaffected by scaling. This point is supported by the work of Boutle et al (2014), in which they analyzed CloudSat data to find the scaling relationship of liquid cloud water. They, too, found that while variability does increase with scale, eventually this relationship plateaus.

[Figure]

[Description for the above figure. Profiles of horizontal variance of qc at each level over the 0900–1200 UTC period for a series of simulations that successively quadruple domain size.]

Secondly, I think it's a significant overreach to say the results are generalizable. The range of number concentration examined did not strongly affect the cloud fraction (much less the cellular organization), such that it's not clear whether a field of open-cell drizzling stratocumulus would respond similarly. Does the profile change for Nc=25 cm-3, an equally realizable concentration at the ENA site? How does the profile change for differing cloud adiabaticity, which would likely be accompanied by a change in the level at which qc variance begins to increase? For differing EIS (or, more specifically, combinations of surface flux magnitude and inversion strength)? For stronger sub-cloud stratification? For a deeper/shallower, moister/drier or cooler/warmer PBL? In short, you need to make a much more forceful argument that these results are generalizable beyond single-layer not-strongly-decoupled stratocumulus decks of approximately the same Nc and LWP as this case. Otherwise, I think you can only say with confidence that LES adequately reproduces the variability of the observations for this specific case. Larger-scale observational studies of column-integrated microphysical variability have showed a broad range of Eq, even in

marine Sc, so saying this one case could be the basis for a global parameterization in the age of big data, routine LES and machine learning seems a bit obsolete.

These are fair questions and criticisms, and we have worked to much more carefully characterize our claim of how general our results are with a new figure and revised paragraph in the Discussion and Conclusions section. First, we made this claim not so much to establish some universal value or profile for IRV or $E_q$ but rather as an explanation for the shape of the profiles. This was not clear in the original submission and we have clarified it. Second, we restrict this claim to cases that include high-cloud-fraction stratocumulus to avoid the issue the reviewer brings up of broken cloud fields (open cells) in clean CCN environments, and the issue of cumulus. The basis of our generalizability claim is the strong constraint on adiabaticity in stratocumulus clouds. To demonstrate the generality of these results, the figure below (now Fig. 11 in the revised manuscript) shows profiles of mean, standard deviation, and IRV of qc for four different published cases of boundary layer cloud taken from the literature (DHARMA LES simulations from Andy Ackerman of the DYCOMS-II RF02 drizzling stratocumulus case and the ATEX cumulus-rising-into-stratocumulus case, and SAMEX LES simulations from co-author Mechem of a CAP–MBL stratocumulus case and a VOCALS case of strongly drizzling stratocumulus in a decoupled boundary layer). We argue that although the absolute magnitudes of the quantities vary across our Fig. 7 and these four cases in the new Fig. 11, broadly speaking the shape of the profiles is similar. Exploring the detailed dependence on the variability on the parameters the reviewer asks about (e.g., adiabaticity, EIS, decoupling) is important but beyond the scope of our study, but the sufficient variation in the four cases shown in the figure above (and Fig. 11) provides some confidence in the reasonableness about our claim that the shape of the profiles is general.

[Figure]

[Description for the above figure. Adding a new figure showing profiles of mean, standard deviation, and IRV of q_c calculated from LES output from four cases of boundary-layer stratocumulus from the literature (DYCOMS–II RF02, Ackerman et al., MWR, 2009), ATEX (Stevens et al., JAS, 2001), CAP–MBL (Rémillard et al., JAMC, 2017), and VOCALS (Mechem et al., JAS, 2012).]

Finally, I suspect the details of the vertical structure of Eq are strongly dependent on the use of a one-moment parameterization. While you didn't get big differences over the range of Nc simulated, a two-moment simulation with varying Nc could yield quite different results, especially at cloud top and base where the variability of number concentration is expected

to be greatest. The benefit of re-running these simulations with bin microphysics is not clear to me; I think the computational resources would be better used incorporating interactive aerosol into a two-moment bulk scheme.

We agree that using 2-moment microphysics that interacts with an activation scheme and interactive aerosol would be a natural next step. That said, we are proposing using bin microphysics in our future steps (already in progress) because a focus of our ongoing work will be to explore how cloud-top entrainment influences the variability at cloud top, in particular as related to homogeneous vs. inhomogeneous mixing regimes. Ideally, the effect of mixing on the details of cloud and precipitation properties will be represented with more fidelity using a bin parameterization.

We agree with the reviewer that some fine details of the vertical structure of $E_q$ likely arise from our use of a single-moment parameterization. However, the profiles from four different cases using bin microphysics in the figure above (now Fig. 11 in the revised manuscript) suggests that the broad shape of our profiles is robust and the results are not especially dependent on the use of a single-moment scheme.

One other significant suggestion: I assume you can diagnose autoconversion/accretion rates directly from the output used to calculate enhancement factors. I think the impact of the figures showing the enhancement factor (e.g., Figs. 8, 10) would be greatly increased by including a profile of domain-mean autoconversion rate. This would give a huge amount of context as to the importance of the enhancement factor on precipitation formation. For example, why is it that you focus on the increase of enhancement factor near cloud top, but not the nearly 2x higher maximum at stratus cloud base? An enhancement factor of 4 vs. 2 doesn't really matter if mean autoconversion rate is 3 orders of magnitude lower at cloud base. At some point in the text you mention that autoconversion is expected to peak in the upper part of cloud, but show it and your point will be made!

We agree that showing a profile of domain-mean autoconversion would help supplement our findings that enhancement factors are lower where autoconversion is more dominant. We have done exactly that and added an autoconversion profile to the enhancement factor plot (Fig. 8). The reviewer also makes a good point that the enhancement factor doesn't really matter all that much near cloud base where the liquid water content and hence autoconversion is so small. This issue is related to the one brought up by Reviewer 1 when asking about vertical dependence in an ESM with only a small number of cloudy points in the vertical. We have added a new paragraph in Sec. 4.3 to discuss these issues.

[Figure]

[Description for the above figure. Panels of revised Fig. 8a that includes the mean profile of autoconversion rate. ]

MINOR COMMENTS

● You use a combination of "sub-grid" and "subgrid" – decide on one hyphenation and use it consistently throughout.

Thanks to the reviewer for pointing out this inconsistency. We have edited the manuscript to only use "subgrid".

● It appears that Fig. 1a was taken directly from Z21. Is this problematic?

Thanks to the reviewer for pointing this out. We acknowledge that Fig. 1a is part of Fig 1 of Z21 and have edited our figure caption to address this:

"Figure is adapted from Fig.1 of Zhang et al. (2021)."

● L268-269: "specifically, that not all the cumulus rise completely into the [Sc] deck" -- is this speculation or did you look into it?

We did look into the cumulus rising into the stratocumulus deck. We looked at many cross sections similar to Figure 6b and 6c and found that there were many instances of small cumulus below ~800m that did not rise into the main stratocumulus layer.

TYPOGRAPHICAL COMMENTS

L6: "variability of the cloud properties that determine the process rate"

Thanks to the reviewer for catching this error. We have corrected the text.

L66: "To account for this bias"

Thanks to the reviewer for catching this error. We have corrected the text.

L74: Remove last instance of "to" in phrase: "more homogeneous to (Lebsock et al…"

Thanks to the reviewer for catching this error. We have corrected the text.

L141: What are "fair-weather" stratocumulus? You chose a pretty heavily drizzling case…

The reviewer is correct that the clouds in our case are strongly drizzling, and in retrospect, "fair-weather stratocumulus" rings oddly to our ears. We have therefore removed "fair-weather."

L145: "Each of these selected legs"

Thanks to the reviewer for catching this error. We have corrected the text.

L147: "which are used in Z21"

Thanks to the reviewer for catching this error. We have corrected the text.

L157: "and peaks at about 1 km for hleg 7"

Thanks to the reviewer for catching this error. We have corrected the text.

L161-L165: This is almost a verbatim repeat of what you said before. Is it really necessary to exactly reproduce it?

We agree that this paragraph is overly repetitive of what we said in the introduction. We have gotten rid of this and revised the end of the previous paragraph to be, "This result indicates that the enhancement factor is vertically dependent and should not simply be treated as a constant. Our simulations aim to provide insights into the mechanisms governing the vertical dependence of $E_q$."

L189: "in situ measurements"

Thanks to the reviewer for catching this error. We have corrected the text.

L217: "The cloud boundaries…"

Thanks to the reviewer for catching this error. We have corrected the text.

L242: I think you mean: "Above this level…"

Thanks to the reviewer for catching this error. We have edited the text as follows:

"peaks within the upper stratocumulus layer"

L262-264: "agrees will with observations and overplotted on…" – the "and" in the middle doesn't make sense to me. This needs to be rewritten but I don't have a specific suggestion.

Thanks to the reviewer for catching this grammatical error and unclear phrasing. We have revised this to the following: "The shape of this profile agrees well with the aircraft observations from Zhang et al. (2021) overplotted on Fig. 7a, which also exhibit an increase in $v_{qc}$ throughout the cloud layer and then decrease closer to cloud top (see also Fig. 1b)."

L267: "likely a consequence of the increase of…"

Thanks to the reviewer for catching this error. We have corrected the text.

L316: either "a large mean value of" or "larger mean values of" – can't combine "a" and plural

Thanks to the reviewer for catching this error. We have corrected the text.

L327: "This allows us to utilize a similar analysis as that used to examine the gradient of…"

Thanks to the reviewer for catching this error. We have corrected the text.

Figure 1b: Is the lower x-axis accurate? This doesn't agree with Z21.

The lower x-axis is correct using our definition of the IRV as mean squared over variance. The x-axis on the plot in Z21 instead uses mean over standard deviation (as noted in their caption), so there is a difference in scale. We have added the following text to the Fig. 1 captions: "Note that the lower x-axis range differs from that in Fig. 4c of Zhang et al. (2021) where their IRV is calculated as mean divided by standard deviation (see their caption), whereas we show mean squared divided by the variance."

Figure 1 caption: replace "KZAR" with "KAZR"

Thanks to the reviewer for catching this error. We have corrected the text.

Figure 7 legend: please note which observed variable you are showing. I assume IRV but it's not unambiguous.

Thanks to the reviewer for catching this error. We have corrected the text.

Figure 7: No indication of what dashed curves mean in legend

In our original drafting of the figure, we tried including the dashed lines representing cumulus in the legend before, but that seemed to make the plot overly cluttered. We therefore chose to keep the dashed lines out of the legend (the same applies to Fig. 8b, where we also chose not to include the dashed lines in the legend). We instead added a description of them in the figure caption.

**Response to Reviewer #3**

The paper uses a small suite of LES simulations to investigate the cloud parameters responsible for an in-cloud enhancement factor used in parameterisations of warm rain processes. The paper is straightforward and succinct. I would recommend for publication following some minor changes.

Main comments

1/ The mean and variance of qc are from in-cloud only values based on the 0.01g/kg threshold. These are then used to derive in-cloud Eq and IRV.

It is important to point out that the final grid mean autoconversion rate depends upon both any in-cloud value of E and a cloud fraction.

So the grid-mean autoconversion will then be E*f(qc, Nc)*Cloudfraction. The case study indicates that E is in the range 1-3, but Cloudfraction can potentially vary by an order of magnitude or more. Both of these need to be considered for coarse models.

We agree with the reviewer that this is an important detail to highlight for ESM/GCMs. In response to the reviewer's comment, we have added the following text (second sentence) to the introduction:

"To account for this bias arising from neglecting subgrid-scale variability in our estimation of < f(x) >, a so-called "enhancement factor" E is introduced so that < f(x) > = E * f(<x>). For processes such as autoconversion, this is conditioned solely in-cloud, so the final grid-mean autoconversion will be dependent on not only E but the cloud fraction as well."

2/ l312 Looking at fig7 in Zhang et al. 2021 it looks like the mode of the qc distribution is below 0.01g/m-3, the threshold used here. The distribution of qc is therefore more like the upper part of a lognormal. In fact, the underlying total humidity distribution (qvapour+qc) is more likely to be normal with the upper tail of it representing the condensed out qc.

If the qc distribution is represented by a truncated distribution how does this impact the results and discussion?

[Figure]

[Description for the above figure. Vertical profile of domain-mean autoconversion rate using locally calculated autoconversion (dashed lines) and autoconversion calculated from the mean cloud water value and then multiplied by the enhancement factor.]

We made the above figure to address reviewer points 1 and 2, which ask about the role of cloud fraction and the sensitivity of our findings to the cloud mixing ratio that we use to determine cloud. In the figure, we show domain-mean profiles of autoconversion two ways. In the dashed lines, we calculate the in-cloud autoconversion point-by-point and then multiply by the cloud fraction to yield the mean autoconversion rate over the entire grid, where both in-cloud autoconversion and cloud fraction are conditioned on two different values of cloud water threshold. We note that there is virtually no difference between the profiles calculated with a threshold of $10^{-3}$ g/m$^{-3}$ and $10^{-2}$ g/m$^{-3}$. The solid lines more closely represent how autoconversion would be calculated in an ESM, where the cloud-conditioned autoconversion is calculated from a mean value of cloud water, and then multiplied by the enhancement factor and cloud fraction calculated with the appropriate threshold value. These lines also lie on top of each other, as well as on the dashed lines which were derived from the point-by-point LES results. The fact that each respective dashed and solid line lie on top of each other is no surprise and is a mathematical necessity given how the

enhancement factor is defined. Most relevantly, the figure concretely demonstrates that the threshold has no real impact on the results, at least for stratocumulus.

3/ l393. Important to caveat this work.

For this case a constant E is definitely a bad idea and overestimates the process rates by large amounts. But...

It is for one case (in part of the diurnal cycle).

It is unclear what this would look like for trade cumulus?

It is unclear what this would look like for Nc significantly greater than 100/cc (e.g. 400/cc).

We acknowledge that this is one case representing typical unbroken marine stratocumulus. Future work will include analyzing different cloud cases from previous studies to identify how the enhancement factor changes for different cloud regimes (ie., stratocumulus, cumulus congestus). Cumulus cloud water profiles are not purely adiabatic, so it is not obvious what the variability and enhancement-factor profiles would be. Looking into varying Nc, we mention in the conclusions that:

"Extending this work using bin microphysics is the object of a future study, which will allow us to consider not only the nonlinearity associated with Nc but also the effect on enhancement factor of homogeneous vs. inhomogeneous mixing regimes (at least down to the limited spatial and temporal scales associated with the model grid and time step) in the entrainment zone."

We currently are running simulations using a bin version of our model as a next step to extend this work. This will allow for a greater understanding of qc-nc variability, but also allow for a varying Nc size. While we have not run a simulation with a significantly large CDNC value, we would assume that the importance of the enhancement factor would decrease due to decreased precipitation and lower overall cloud water variability.

minor points

l38. 'many' might be a slight exaggeration - only one is cited.

We added a few more references to address this wording issue.

l52. You do introduce it later on but it might be worth a sentence here to note that accretion is also important for precipitation production. Furthermore it complicates the situation further by having to deal with the collocation of 'rain' and 'cloud' species. This will be even more important for cumulus.

In response to the reviewer's comment, we have emphasized the importance of accretion in the precipitation process and also clarified that because accretion is only weakly nonlinear, enhancement factors related to accretion will be small. We have added the following text: "Although accretion is critically important in the growth of precipitation drops, its nonlinearity is rather weak ($\sim(qc*qr)^{1.15}$ in Khairoutdinov and Kogan 2000), so any associated accretion enhancement factor will be small." We agree that representing the precipitation process for subgrid-scale cumulus, especially the collocation of the relevant microphysical variables, is a huge challenge that we are not addressing.

l146. There is an opportunity here to also analyse the data as 10km and 1km legs to show the scale dependence. It will be of interest to see if E is significantly different to 1 at 1km

scales.

[Description for the above figure (repeat of figure also shown in the response to Reviewer 2). Profiles of horizontal variance of qc at each level over the 0900–1200 UTC period for a series of simulations that successively quadruple domain size.]

Reviewer 2 also had a question about scale. We acknowledge the reviewer's interest in analyzing the data to show scale dependence, but we mainly focus on the scale needed to compare to aircraft observations as opposed to overall scale dependence. In the spirit of the reviewers' questions, we performed a number of additional simulations and calculated the variance at a number of scales (see the figure above). We found that variance increased up

to a scale of ~4.5 km (128x128) and that scales beyond that did not introduce additional variability.

[Figure]

[Description for the above figure. Vertical profiles of the enhancement factor Eq. Solid lines represent total values while dashed lines represent values correlating to only cumulus clouds. Purple, dark blue, and teal lines represent simulations with Nc values of 100, 75, and 50 cm⁻³, respectively, all run over the smaller 256x256 (8.96x8.96 km²) grid. The black lines correspond to the control run using the larger 30x30 km² domain. Note that the black and dark blue lines practically overlie each other. ]

As for analyzing the data at a scale of 10km, our droplet concentration study actually shows this scale for the enhancement factor as those model simulations used a domain of 256x256 points, or 8.96km by 8.96km. As previously mentioned, since the qc variance is nearly identical between scales of ~10km to ~30km, the overall enhancement factor is also identical for simulations using a nc value of 75cm-3 (dark blue line is 256x256, black is 864x864). This additional profile corresponding to the larger (control) domain was added to Fig. 10.

l151. Where does the measured qc and nc come from? FSSP, CDP, King probe? I appreciate that all the observational details are not too important for this, but does the observed qc include all liquid droplets including ones that might be considered rain by the autoconversion parameterisation?

The measured qc and nc comes from the fast cloud droplet probe of the G1 aircraft as stated in Zhang et. al. 2019:

"The size distribution of cloud droplets, and the corresponding $qc$ and $Nc$ are obtained from the fast cloud droplet probe (FCDP) measurement. he FCDP measures the concentration and size of cloud of size around 3 μm *(Lance et al., 2010; SPEC, 2019).* Following previous studies (Wood, 2005a), Z21 adopted r* = 20 μm as the threshold to separate cloud droplets from drizzle drops; i.e., drops with r < r* are considered cloud droplets. After the separation, the qc and Nc are derived from the FCDP droplet size distribution measurements. In other words, only the cloud-mode (with r < r*) droplet from the FCDP is used in Z21. As an evaluation, Z21 compared the FCDP-derived qc results with the direct measurements of qc from the multielement water content system (WCM-2000; Matthews and Mei, 2017) also flown during the ACE-ENA and found a reasonable agreement (e.g., biases within 20 %). Z21 also performed a few sensitivity tests in which we perturbed the value of r* from 15 up to 50 μm. The perturbation shows little impact on the results"

We have referred the reader to Zhang et al. (2019) for these details.

l156 fig1b. Could add on points for 10km and 1km legs.

A more thorough exploration of the scale dependence of the variability metrics (IRV, E, etc.) is definitely worthwhile but is somewhat beyond the scope of this study. At least as far as these figures are concerned, it would also add a great deal of clutter. We will explore scale dependence more deeply in our follow-on paper that accounts for variability of both qc and CDNC.

l159. It would be useful to define how E is derived before this point in the text.

We acknowledge the reviewer's comment that it is useful to have a definition of E before it is discussed. We have edited the text as follows:

"E is defined as E=< f(qc)> /f(< qc>), where f is the autoconversion function. Since the value of E for autoconversion due…"

l215. 0.01g/kg : is this the same threshold as the aircraft observations?

The aircraft observations also use a 0.01 g/kg threshold.

l301. Does the good agreement between the model profiles of IRV qc, Eq  and observed profile mean that there is no need to worry about independently varying Nc?

No, we believe that studying variability Nc in addition to qc will not only make our simulations more representative of the actual environment, but it will also provide more information on how the enhancement factor changes in the entrainment zone. Zhang et al. (2021) established that the covariability of qc and Nc has important implications for the enhancement factor. While the effect of varying Nc may not be as pronounced in the lower 2/3 of the stratocumulus layer, Nc varies much more near cloud top due to entrainment. Independently varying Nc will allow us to

examine the effects of homogeneous vs. inhomogeneous mixing regimes near cloud top on the enhancement factor.

However, most of the current GCMs do not consider the sub-grid variation of Nc or its correlation with qc. Therefore, even though there is strong evidence that the sub-grid variations of qc and Nc both influence the Eq and the auto-conversion rate computation in GCM, this influence is not represented in the GCM yet. We could not say that there is "no need to worry about independently varying Nc," but at the current stage, we also need to understand the subgrid variability of qc better.

l381. Give the range for this case.

We have added the ranges for IRV and $E_q$ to this conclusion:

"Both inverse relative variance $v_{qc}$ and enhancement factor Eq vary considerably in the vertical. $V_{qc}$ ranges from 0 to nearly 9, while $E_q$ ranges from close to 1 to 3.70."

---

## Referee Report (RR1)

Second review, Covert et al., acp-2021-656

The authors' response to my and the other reviewers' comments have greatly improved the manuscript and I recommend it for publication pending a couple extremely minor technical corrections (see final paragraph).

I am still of the opinion that the shape of the enhancement factor profile is only "general" under conditions of "idealized single layer stratocumulus with a well-defined inversion," as this is what gives rise to the adiabaticity constraint discussed by the authors (e.g., line 397 of revised manuscript), but I appreciate the effort to include Fig. 11. It lends great confidence to the idea that there is a consistent (if perhaps not "universal") structure to the profile of $E_q$.

I also very much appreciate the addition of the autoconversion (Au) rate curve to Fig. 8 and the discussion in lines 310-313 and 439-445 of the implications of lower $E_q$ coinciding with maximum Au rate near cloud top. I think this is one of the most impactful findings in terms of GCM microphysics development, and I look forward to the authors' future work with a two-moment scheme to evaluate whether variability of N in the entrainment zone augments or dampens $E_q$. As the authors state, this is likely to be highly dependent on mixing assumptions.

Lastly, I must admit that I made a mistake in suggesting a reference in my first review. I intended to suggest the authors cite Lebo et al. 2015 (https://doi.org/10.1175/JAMC-D-15-0066.1) instead of Lebo and Feingold 2014. One other minor recommendation is to explicitly state the type of microphysics scheme used to produce the results shown in Fig. 11 – the author response states that these simulations are all with bin microphysics, but this is not reflected in the manuscript.

Mikael Witte